# Ultrafast neural sampling with spiking nanolasers

**Ivan K. Boikov** [1] ✉, **Alfredo de Rossi** [1] & **Mihai A. Petrovici** [2]

Owing to their significant advantages in terms of bandwidth, power efficiency, and latency, optical neuromorphic systems have arisen as interesting alternatives to digital electronic devices. Recently, photonic crystal nanolasers with excitable behavior were first demonstrated. Depending on the pumping strength, they emit short optical pulses – spikes – at various intervals on a nanosecond timescale. In this theoretical work, we show how networks of such photonic spiking neurons can be used for Bayesian inference through sampling from learned probability distributions. We provide a detailed derivation of translation rules from conventional sampling networks, such as Boltzmann machines, to photonic spiking networks and demonstrate their functionality across a range of generative tasks. Finally, we provide estimates of processing speed and power consumption, for which we expect improvements of several orders of magnitude over current state-of-the-art neuromorphic systems.

In the overwhelming majority of artificial neural networks (ANNs) in use today, neuronal outputs are determined by smooth and continuous activation functions, and their values are updated synchronously for multiple neurons within a network. In contrast, biological neurons, especially in the mammalian cortex, communicate asynchronously with short, stereotypical pulses called spikes. While the non-differentiability of such spike-based codes indeed makes them more difficult to train using error backpropagation and therefore less attractive for conventional deep learning (however, see refs. [1–3]), they possess several properties that are extremely relevant in physical neuronal networks, biological and artificial alike. First, they can be much more efficient than time-continuous or rate codes: as information becomes implicitly encoded in the timing of the spikes, rather than the value of neuronal outputs, a spike code can save both energy and bandwidth. Thus, spiking neurons have become the de facto standard model for neuromorphic systems, both purely digital[4–6] and mixed-signal[7–10]. Second, due to the all-or-nothing nature of spikes, the transmission of information is significantly less prone to disruption by noise. This makes spiking neural networks particularly attractive for mixed-signal devices, whose analog components (neurons and synapses) invariably introduce spatio-temporal noise into the network dynamics.

By associating neuronal spikes with binary states, one can link spiking neural networks to Boltzmann machines (BMs)[11–13] (but see also ref. [14] for an alternative implementation). This machine learning model forms the basis for powerful variants such as deep belief networks[15] and autoencoders[16], which have been used for applications ranging from acoustic modeling[17] to medical diagnostics[18] and quantum tomography[19]. By moving to the spiking domain, one can harness the speed and efficiency of neuromorphic substrates for such applications. Indeed, first demonstrations of such spike-based sampling networks on highly accelerated neuromorphic hardware have already shown promising results[20–22].

However, despite the impressive results of electric and electronic networks, electrical interconnects between computing elements limit bandwidth, latency, and energy efficiency[23]. In contrast, photonic transmission of information offers significant advantages in terms of loss, bandwidth, and computing speed. Not only is the issue of Joule heating alleviated, but the improved fan-in/-out by wavelength multiplexing also facilitates large-scale connectivity. The initial motivation of improving interconnections between digital electronic computing elements has recently extended to actual photonic computing, in particular, implementing matrix-vector multiplication. Two studies have recently and independently introduced a 64 × 64 photonic matrix demonstrating exceptional computing speed and latencies[24,25]. Current research on photonic neural networks implements artificial neurons similar to those in machine learning, which are deterministic and have a continuous activation function[26].

¹Thales Research & Technology, Palaiseau, France. ²Department of Physiology, University of Bern, Bern, Switzerland. ✉e-mail: mail@ikboikov.net

In this work, a photonic equivalent of a neuron is a laser with an excitable response[27–33]; interaction between such neurons with highly accelerated intrinsic dynamics is being developed[34]. Among a variety of possible implementations, a semiconductor laser with a saturable absorber (SA) is of particular interest here[35] because of the strong analogy with the leaky integrate-and-fire (LIF) neuron model[36].

Semiconductor lasers can be miniaturized and integrated in a photonic circuit and need tiny amounts of power (about 10 $\mu$W)[37], while, by trading higher levels of current (100 $\mu$W), nearly 14% wall-plug efficiency directly into a silicon photonic circuit has been demonstrated[38]. These are essential requirements for advancing the neuromorphic state of the art. Very recently, this technology was shown to produce an experimentally observed excitable response by adding a resonator section acting as an SA[39], thus paving the way for efficient integrated photonic spiking neurons (PSNs). The combination of a pulsed, stochastic response followed by a pronounced refractory period allows the implementation of neural sampling as a means of approximating the Glauber dynamics found in BMs. In contrast to most photonic neural networks found in literature, such networks operate on probability distributions rather than signals, which fills an important niche of processing non-deterministic and partial inputs. In this work, we demonstrate the capability of such PSN networks to implement BMs through spike-based sampling.

## Results

### Preliminaries

A BM is a stochastic neural network composed of binary neurons connected by a symmetric weight matrix. All possible states of the network can be described with a set of vectors $\mathbf{z}$, where $z_k \in \{0, 1\}$, i.e., not active or active. The probability of the BM being in a particular state $\mathbf{z}$ is $p(\mathbf{z}) \propto \exp\left[-E(\mathbf{z})/k_B T\right]$, where $k_B$ is the Boltzmann constant, $T$ is the ensemble (Boltzmann) temperature, and $E(\mathbf{z})$ is the energy of this state

$$E(\mathbf{z}) = -\sum_{k<j} W_{kj} z_k z_j - \sum_k b_k z_k, \tag{1}$$

where $b_k$ are the neuronal biases, and $\mathbf{W}$ is a symmetric coupling matrix that represents synaptic weights. Since the influence of the temperature $k_B T$ can be absorbed into the weights and biases, we can disregard the units, assume $k_B T = 1$, and omit this term for brevity from here on.

In general, a BM can be used for Bayesian inference in previously learned probability spaces. First, biases and weights are adapted such that $p(\mathbf{z})$ follows a particular distribution determined by the training data. Then, an input (evidence or observation) is provided by changing biases, and the output (the result of the inference problem) is simply obtained by measuring the resulting changes to $p(\mathbf{z})$. For example, one can train a BM to follow a distribution over natural images and their labels. Afterwards, when provided with a partial image (by bias-clamping the neurons representing the corresponding pixels to their respective values), the BM will generate states of the unobserved neurons that correspond to plausible completions of the partial image, along with the corresponding labels.

However, the state space of $N$ binary neurons has a size of $2^N$, which renders the computation of the probability distribution intractable even for modest networks. Sampling can be a viable alternative: a histogram of samples approximates the true probability distribution, and the accuracy increases with more samples drawn. Fast sampling is therefore preferable, for which the nature of the hardware implementation matters a lot. Here, we consider an implementation with spiking neurons, as their dynamics lend themselves to ultrafast and energy-efficient implementation with nanolasers, as discussed further below.

To start, consider a time-continuous counterpart to a classical BM neuron. Whenever a neuron fires a spike, it enters a refractory period

of some duration $\tau$ during which it cannot spike again. Thus, a spike can be interpreted as representing the onset of an active state

$$z_k(t) = 1 \iff \text{neuron has fired in } (t - \tau; t], \tag{2}$$

with $z_k(t) = 0$ otherwise. Conversely, a network of $N$ spiking neurons can then serve to represent the full state $\mathbf{z}(t) \in \{0, 1\}^N$.

In spike-based sampling, a spiking neural network approximates $p(\mathbf{z})$ through a time-continuous analogon of Gibbs sampling[11], a sampling algorithm where components $z_i(n)$ of a current sample $\mathbf{z}(n)$ are updated iteratively using the conditional probability distribution: $z_i(n+1) \sim p(z_i | z_k(n) \; \forall_{k \neq i})$. A particularly interesting variant builds on LIF sampling (LIFS) neurons[12,13], as they represent a de-facto standard model across the vast majority of neuromorphic platforms. In LIFS, the required stochasticity is generated by adding (or exploiting pre-existing) noise on the neuronal membranes, which consequently follow the Ornstein-Uhlenbeck process

$$\tau_m du_k = (b - u)dt + \sum_j \sum_{t_j^{spike}} W_{kj}\kappa\left(t - t_j^{spike}\right) + \sigma_W dW_t \tag{3}$$

with an autocorrelation determined by the neuronal membrane time constant $\tau_m$ ($dW_t$ represents a Wiener process scaled by a noise amplitude $\sigma_W$). The interaction between neurons is mediated by an additive postsynaptic potential (PSP) kernel $\kappa$ that is triggered by an incoming spike at time $t^{spike}$.

### Spiking nanolasers

The PSNs discussed in this work are semiconductor lasers composed of two sections: a gain section and an SA. The two sections form a single resonator and are therefore spanned by a single laser mode (Fig. 1a). The gain section is pumped to reach local electron population inversion and stimulated emission of photons. The SA is not pumped, such that here absorption always dominates, yet saturates as the flux of absorbed photons increases. As the pump rate increases beyond a certain threshold, the absorption is overtaken by the stimulated emission; akin to negative differential resistance in electronic oscillators, this introduces a positive feedback which initiates the emission of a pulse. In turn, this depletes the population of excited electron-hole pairs, hence the gain, and the laser is shut off. Therefore, the emission of a new pulse immediately following a previous one is strongly suppressed. After some time, the pumping re-establishes the initial gain, and the nanolaser can spike again. This process (Fig. 1b) bears a strong resemblance to the generation of action potentials and the subsequent refractoriness found in biological neurons, as described by the Hodgkin-Huxley model[40].

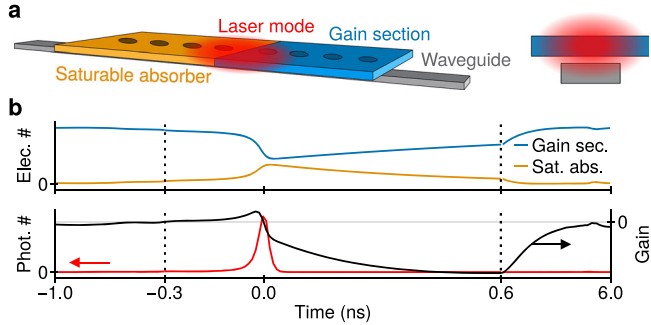

**Fig. 1 | Spiking nanolasers. a** Schematic of a PSN coupled to a silicon waveguide seen from above (left) and the midsection of the structure (right). Here, the laser is a nanobeam photonic crystal; its modes are standing-wave and optical spikes are coupled out to the waveguide in both directions. **b** Optical spike generation. Dashed lines separate sections with different scaling of the time axis.

The spiking dynamics in a semiconductor laser with an SA is described by the Yamada model[35,41], which ignores spontaneous emission and therefore assumes a perfectly deterministic response. Here, we consider a laser with a single optical mode; its field is distributed over a volume comparable to $\lambda^3$, where $\lambda$ is the wavelength. Moreover, the active region, where electron and hole pairs are created, is even smaller. Under these conditions, spontaneous emission is not negligible, as the fraction of it going into the mode – the spontaneous emission factor – is between 0.1 and 1.0, whereas in macroscopic semiconductor lasers it is $10^{-4}$ or less[42]. Therefore, in nanolasers, the average number of photons is much smaller, and the relative noise due to the granularity is much larger, which disrupts the otherwise deterministic spiking at regular intervals controlled by pumping strength.

A rigorous description of noise requires a quantum mechanical formalism, which is exceedingly complicated for semiconductor systems. Therefore, the semiconductor laser is often approximated as a homogeneously broadened two-level system, as intraband scattering is fast enough such that electrons and holes are in thermal equilibrium[43,44]. The light-matter interaction within a semiconductor can therefore be described as a collection of dipoles interacting with the same optical mode. With some approximations, such as fast dephasing of the polarization with respect to the damping rate of carriers and photons, the quantum mechanical descriptions lead to rate equations[45,46].

The rate equations describe the evolution in time of the populations of excited dipoles $n_e$, and photons $S$ in a cavity mode due to multiple processes. Stimulated emission and absorption are described by a term $\gamma_r(2n_e - n_0)$, where $n_0$ is the total number of dipoles, and $\gamma_r$ is the radiative transition rate. Since $n_e \leq n_0$, this term is limited to $\gamma_r n_0$; this condition is ensured by an optical pumping term $\gamma_p(n_0 - n_e)$, where $\gamma_p$ is the pumping rate and $n_0 - n_e$ represents pumping saturation. Spontaneous emission and photon damping are described by $\gamma_r n_e$ and $\gamma S$.

The stochastic equations are formed by including the Langevin forces $F_i(t)$, which are random variables with zero mean and auto-/cross-correlation strengths $\langle F_i(t)F_j(t)\rangle = 2D_{ij}$. The Langevin forces are calculated consistently with the rate equations[46] based on the McCumber noise model[47] (see Sec. "Nanolaser noise model" in the Methods). Here, we extend the model by including two separate sections, one providing amplification with an excited population $n_e$, and another one representing the SA, denoted with the suffix $a$, with an excited population $n_a$, both interacting with the same mode:

$$\dot{S} = GS + \gamma_r n_e + \gamma_{r,a} n_a + F_S(t),$$
$$\dot{n}_e = -\gamma_r(2n_e - n_0)S - \gamma_t n_e + \gamma_p(n_0 - n_e) + F_e(t), \quad (4)$$
$$\dot{n}_a = -\gamma_{r,a}(2n_a - n_{0,a})S - \gamma_{t,a}n_a + F_a(t),$$

where $G = \gamma_r(2n_e - n_0) + \gamma_{r,a}(2n_a - n_{0,a}) - \gamma$ is the gain including photon damping, and other parameters are defined in Supplementary Table 1. For clarity, we normalize $\gamma_p$ by the threshold pumping strength in the absence of noise $\gamma_p^{thr}$ (see Sec. "Nanolaser with quantum wells" in the Methods) which we later refer to as the threshold. It is important to note that with noise, spike emission can also occur with $\gamma_p < \gamma_p^{thr}$.

Figure 2 a shows a time trace of the membrane potential of a single LIFS neuron with parameters from ref. 13 and Gaussian noise. This serves as a reference for the behavior of PSNs, which we discuss below. In Fig. 2b, we show time traces of PSN gain with pumping far from and close to the spiking threshold, respectively. In this system, the gain is a stochastic variable as it depends on stochastic electron densities. As a result, when a PSN is not refractory, the gain undergoes a random walk as shown in Fig. 2c, similarly to the LIFS membrane potential. The inter-spike interval distribution, gain autocorrelation, and probability density function of a PSN are in Fig. 2d, e, and f, respectively. These are close to the corresponding properties of LIFS neurons, with small deviations explained by the additional nonlinear terms of the PSN state equations.

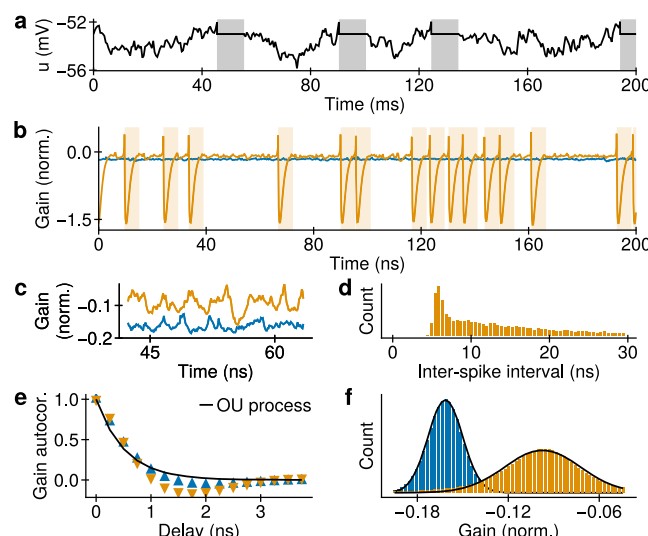

**Fig. 2 | PSNs emulate LIFS neurons. a** Membrane potential of a LIFS neuron. Shading represents the active state after a spike emission. Below: properties of a PSN far from (blue, $\gamma_p = 0.955\gamma_p^{thr}$) and close to (orange, $\gamma_p = 0.995\gamma_p^{thr}$) the spiking threshold. **b** Gain time traces. After a spike is emitted, the gain is reduced drastically, and the PSN is considered active for a time shown with shading. **c** Gain time traces between spike emissions. **d** Inter-spike interval histogram. **e** Autocorrelation of the gain. For each spike at $t_s$, the interval ($t_s - 0.5\tau$; $t_s + \tau$) was omitted from the analysis to exclude the highly nonlinear regime dynamics of the spiking process. The case far from the threshold is fitted with an Ornstein-Uhlenbeck process, as also obeyed by the free membrane potential of LIFS neurons. **f** Histograms of gain values between spike emissions and fit with normal distributions (black lines).

It has been pointed out that the granularity of light challenges numerical integration based on the Langevin forces[48–50]. Therefore, we further cross-checked the results by implementing a rigorous discrete PSN model, as proposed in ref. 49 and concluded that both methods lead to the same statistical properties of the PSN (see Sec. "Discrete nanolaser model" in the Methods).

## Networks of spiking nanolasers

While many different implementations of our proposed networks can be conceived, we discuss several ideas in the following. Interference-based interaction between PSNs is challenging: their bistability, combined with a potentially large number, will complicate the necessary resonance alignment. For this reason, in this work we assume that PSNs are incoherent, and their interaction is mediated by photodiodes. Spikes are extracted from a PSN by coupling to a waveguide. All-to-all connection weights can be implemented using a photonic matrix (see Fig. 3c), a photonic integrated circuit developed to implement matrix-vector multiplication[24,25]; the use of this technology here assumes incoherent superposition, as nanolasers may emit at different wavelengths, yet this still leaves a margin for setting the weights to their desired values. Wavelength diversity also allows the combined use of optical crossbars (see Fig. 3d) based on wavelength division multiplexing[51]. In this respect, potential limitations due to the intentionally reduced interference and to the fact that the detector measures a positive quantity may need to be managed. In fact, in BMs, weights can be negative as well. A possible solution is the use of balanced photodetectors[52], which requires an $N \times 2N$ coupling matrix for $N$ PSNs.

As a result, optical spikes are converted into an electrical current that changes the pumping strength by

$$\Delta\gamma_{p,k}(t) = \sum_j \kappa_{kj} \int LPF(t - t^*)S_j(t^*)dt^*, \quad (5)$$

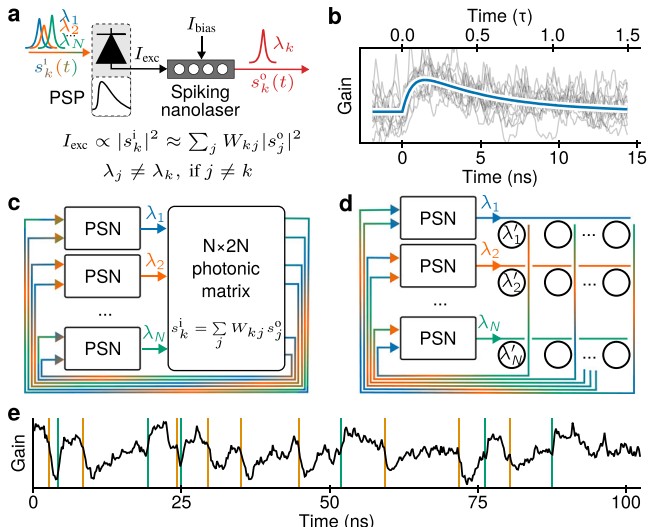

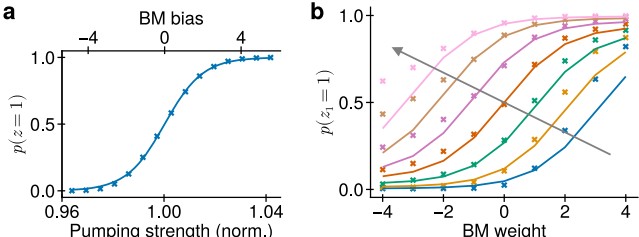

**Fig. 3 | Setup of a PSN network. a** Possible implementation of a PSN with a photodetector and a nanolaser for incoherent excitation. An optical signal $s^{(i)}(t)$ composed of optical spikes $s_j^{(i)}(t)$ at various wavelengths $\lambda_j$ (shown with various colors) is absorbed by a photodetector. Limited electrical circuit bandwidth leads to filtering of the spike, resulting in an almost alpha-shaped PSP. The sign of $I_{exc}$ impacts the emission of a spike. **b** Change of the nanolaser gain upon a reception of a spike (i.e., a PSP). The blue line shows an alpha-shaped fit. **c** Possible interconnection of nanolasers using a $N \times 2N$ photonic matrix that acts as a tunable broadband scatterer. Here, 2N outputs are required for balanced photodetectors that provide positive and negative connections between PSNs. **d** Same, with a waveguide crossbar array and microrings as weighting elements. Here, each wavelength is selectively weighted, making tuning easier than in (**c**). **e** Gain time trace of a PSN affected by incoming spikes (vertical lines) from two other PSNs: one is connected with a positive weight (excitatory, green) and the other with a negative one (inhibitory, orange).

where $\kappa_{kj}$ represents a coupling strength incorporating relevant factors such as the coupling of a PSN to a waveguide and the photodiode responsivity, and $LPF(\Delta t)$ is the impulse response function of the photodiodes that we assume to be a low-pass filter with a timescale $\tau_U$ due to their limited bandwidth. The change of current induces a change of gain shown in Fig. 3b; the change can be positive (excitatory) or negative (inhibitory), as shown in Fig. 3e. In the next section, we propose that the gain plays the role of the membrane potential in a PSN. Therefore, Eq. (5) describes the PSP; its shape is largely defined by the bandwidth of the electrical scheme (Fig. 3a,b).

This implementation of connections induces an almost alpha-shaped interaction kernel $(t/\tau)\exp(-t/\tau)$, again similar to typical interaction kernels in both biological and abstract (LIF) neuronal networks. For optimal performance of a network, the timescale of $\Delta\gamma_{p,k}(t)$ can require optimization. Here, it depends on $\tau_U$, which we assume to be tunable, as discussed further below.

**Membrane potential of spiking nanolasers**

Consider a network of nanolasers and its $k$-th PSN is not currently spiking, i.e., its gain does a random walk (see Fig. 2c). Then, Eq. (4) is approximated by the following stochastic equation for the gain (see Sec. "*Simplified gain equation*" in the Methods):

$$dG_k \approx \left[-(G_k - G_{p,k})/\tau_G + 2\gamma_r n_0 \Delta\gamma_{p,k}\right]dt + \sigma_G dW_k, \quad (6)$$

where $G_p$ is a drift term depending on the pumping strength, and the second term corresponds to the PSN interaction (see Eq. (5)). This equation is identical to that of the membrane potential of an LIFS neuron (see Eq. (3)) with two assumptions. First, $n_a \ll n_e$, which holds during non-spiking with moderate pumping strength. Second, changes

**Fig. 4 | Translation of Boltzmann parameters to PSN parameters. a** Spiking nanolaser activation function (crosses) with a logistic function fit (line). **b** Impact of connection weight on activation of a receiving neuron. Lines and crosses show activation of a BM neuron and a nanolaser, respectively. Each color corresponds to a receiving neuron bias $b_1 = -3, -2, ...3$ increasing along the arrow.

of $n_e$ due to incoming spikes need to be small, otherwise the interaction between PSNs deviates from linearity (see Sec. "*Simplified gain equation*" in the Methods). Based on the formal equivalence between the two equations, we propose to use the gain $G$ as the membrane potential of a PSN:

$$u_k(t) = (G_k(t) - G_0)/\partial_u G, \quad (7)$$

where $G_0$ is the gain for which $p(z = 1) = 0.5$ (which requires the yet unknown refractory period $\tau$), and $\partial_u G$ is a proportionality coefficient. Therefore, we must to find these three parameters to relate the membrane potentials of LIFS neurons and PSNs.

Consider a single LIFS neuron with a logistic activation function $p(z = 1|\bar{u}) = \sigma(\bar{u})$. We expect a similar behavior from a single PSN, where $\bar{G} = G_p$ (see Eq. (6)). In Fig. 4a we sweep the pumping strength $G_p$; for each $G_p$, we solve Eq. (4) for a sufficiently long time and estimate the activation function using Eq. (2) and the law of large numbers:

$$p(z = 1|G_p) \approx \bar{z}(G_p), \quad (8)$$

where $\bar{z}$ is an average PSN activation. Here and throughout the article, the PSN activation is discretized with a timestep $\tau$ before processing, i.e., $\bar{z} = \langle z(m \cdot \tau) \rangle_m$. The goal is then to ensure that

$$\bar{z}(G_p) \approx \sigma\left[\left(G_p - G_0\right)/\partial_u G\right], \quad (9)$$

which is an optimization problem for the three variables $\tau$, $G_0$ and $\partial_u G$. With it solved, we find that the PSN activation function matches the logistic activation function of LIFS neurons well. Here, $\tau = 11$ ns, which corresponds to a sampling rate of approximately 0.1 GHz. Based on the similarities between the expressions for membrane potentials of LIFS neurons (Eq. (3)) and PSNs (Eq. (6)) we find the following translation rules between the bias and the pumping strength:

$$b_k = (G_{p,k} - G_0)/\partial_u G. \quad (10)$$

Connections between LIFS neurons induce a change of their membrane potentials according to Eq. (3). Consider two LIFS neurons connected unidirectionally, i.e., only $W_{12} \neq 0$, with $b_2 = 2$. This way, the second neuron acts as the source of spikes, while the first is the recipient. Sweeping $b_1$ and $W_{12}$ we obtain a set of curves $p(z_1 = 1|b_1, W_{12})$ shown with lines in Fig. 4b. We now aim for a similar behavior in PSNs.

We convert $b_1$ and $b_2$ to PSN pumping strength according to Eq. (10). We then sweep $\kappa_{12}$ to obtain a set of curves $p(z_1 = 1|G_{p,1}(b_1), \kappa_{12})$. The goal is to find a proportionality coefficient $\xi$ such that

$$p(z_1 = 1|G_{p,1}(b_1), \xi\kappa_{12}) \approx p(z_1 = 1|b_1, W_{12}). \quad (11)$$

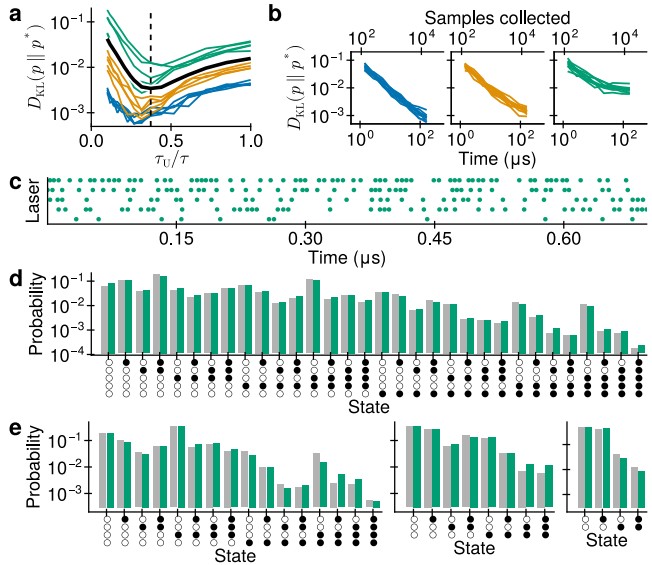

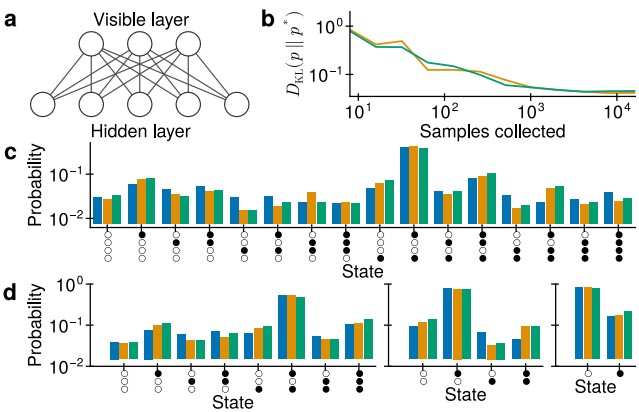

**Fig. 5 | Sampling from random Boltzmann distributions with PSN networks.**
**a** Optimization of the photodiode timescale: $\tau_U$ is swept and sampling performance computed for PSN networks with maximum weights of 0.6 (blue), 1.2 (orange) and 2.4 (green). The solid black line is the mean $D_{KL}$ for each $\tau_U$. The dashed line shows $\tau_U = 0.37\tau$. **b** Convergence of sampling from 10 random Boltzmann distributions with weights up to 0.6, 1.2 and 2.4 from left to right. **c** Spike raster during sampling from a Boltzmann distribution with weights up to 2.4. **d** Sampling result for a Boltzmann distribution in (c). Gray: analytical distribution, green: sampling result. **e** Sampling from conditional distributions. From left to right: $p(z_{1345}|z_2 = [1])$, $p(z_{245}|z_{13} = [1, 0])$ and $p(z_{12}|z_{345} = [1, 1, 1])$. Colors match (**d**).

**Fig. 6 | Sampling from an arbitrary distribution with a network of PSNs.**
**a** Architecture of the implemented PSN network. Each line represents a pair of symmetric synaptic connections. **b** Convergence of sampling with an RBM (orange) and a PSN network (green). **c** Comparison of the target distribution (blue) to the distributions sampled by the RBM and the PSN network. **d** Sampling from conditional distributions. From left to right: $p(z_{134}|z_2 = [0])$, $p(z_{24}|z_{13} = [1, 0])$ and $p(z_3|z_{124} = [1, 0, 1])$. Colors match (**c**).

The result is given in Fig. 4b. We find that for limited biases and weights, PSNs replicate the behavior of LIFS neurons well. Based on Eq. (11), we find the translation rule for weights:

$$W_{kj} = \xi \kappa_{kj}. \tag{12}$$

To conclude, we have found that dynamics of PSNs and their interaction imitate that of neurons of a BM. In the following sections, we investigate the accuracy of the imitation by considering a series of tasks of increasing complexity.

## Optical sampling from Boltzmann distributions

To investigate the accuracy of sampling with PSNs, we first consider sampling from predefined Boltzmann distributions over a small set of binary random variables. Following[13], biases and weights were drawn from Beta distributions, $b_k \sim 1.2(\mathcal{B}(0.5, 0.5) - 0.5)$ and $W_{kj} \sim \beta(\mathcal{B}(0.5, 0.5) - 0.5)$, where $\beta$ controls the range of weights and is either 0.6, 1.2, or 2.4. The generated biases and weights are translated to PSN parameters using Eqs. 10 and 12. The sampled distributions $p$ are compared to the target distributions $p^*$ by means of the Kullback-Leibler divergence $D_{KL}(p\|p^*)$.

First, we optimize the PSP timescale controlled by $\tau_U$. On one hand, $\tau_U$ must be small enough such that a PSP does not last longer than the refractory period $\tau$. On the other hand, too short $\tau_U$ will make the PSP short, but strong, which can break the operating regime assumptions (see Sec. "*Simplified gain equation*" in the Methods). In LIFS neurons, the PSP timescale is ideally slightly shorter than the refractory period[13]; we use this as a starting point. In Fig. 5a, we sweep $\tau_U$ and track the sampling accuracy for all considered $\beta$. We find that the optimal $\tau_U$ is approximately $0.37\tau$. Figure 3b shows a PSP with such a timescale, and indeed, the PSP becomes negligible after $t = \tau$.

We proceed to sample from sets of 10 random Boltzmann distributions for different $\beta$. Figure 5b shows the convergence of the sampling procedure towards the target distribution. We find that

within each set, the convergence is almost identical. In Fig. 5d, we compare a distribution of samples to the exact distribution for $\beta = 2.4$, and we note a very close match. Figure 5c shows a raster of spikes during this sampling.

For the next test, we split the five neurons in two arbitrary groups, Y and X. The first group is clamped to an arbitrarily chosen state **y**, and the neurons in the second group are free; their probability distribution is the conditional probability distribution $p(X|Y = \mathbf{y})$. In PSNs, we clamp the state by significantly reducing or increasing the pumping strength. During sampling, we collect samples of the free neurons and this way, estimate $p(X|Y = \mathbf{y})$. In other words, the PSN network samples from the conditional probability distribution of the BM. Figure 5e shows the close match between the correct conditionals and those sampled with our PSNs. We therefore conclude that a network of PSNs can sample accurately from a wide range of Boltzmann distributions and their conditionals over small state spaces.

## Optical sampling from arbitrary distributions

Boltzmann distributions are only a subset of all possible probability distributions over binary variables. However, the fully visible sampling networks described above can be extended by adding hidden neurons that are not observed during sampling. Consequently, the probability distribution of the visible layer becomes a marginal distribution over the full state space, which can, in principle, take any shape, given a large enough hidden space. To simplify training and improve convergence, a hierarchical network structure is preferable, with no horizontal connections within individual layers[15]. Here, we emulate such a two-layer restricted Boltzmann machine (RBM) with an equivalently structured PSN network (see Fig. 6a). Given enough hidden neurons, an RBM can sample from any distribution with arbitrary precision[53]. Consequently, by implementing such an RBM with PSNs, optical sampling from arbitrary distributions can be achieved.

Here, we choose a target probability distribution $p^*(\mathbf{z})$ with 4 binary variables, which can be fully defined by $2^4 = 16$ probabilities, which we generate by sampling 16 numbers from the inverse continuous uniform distribution over [0, 1] (whose probability density is $f(x) = x^{-2}$ for $x \in [1, \infty]$ and zero elsewhere) and normalizing them such that their sum is unity. The probability distribution is shown in Fig. 6c.

Using the contrastive divergence algorithm (see Sec. "*Contrastive divergence*" in the Methods), we train an RBM with 4 visible and 10 hidden neurons. Its sampled distribution over visible neurons $p_{RBM}(\mathbf{z})$ is shown in Fig. 6c.

The parameters of the trained RBM were then transferred to a network of 14 PSNs. Its sampled distribution over visible neurons $p_{PSN}(\mathbf{z})$ is also shown in Fig. 6c. We find that the accuracy of the PSN network is very close to that of the RBM, and sampling from the target distribution $p^*(\mathbf{z})$ and several conditionals shown in Fig. 6d is correspondingly accurate.

## Optical probabilistic inference

In Bayesian statistics, a probability represents the degree of belief in an event. Bayesian inference is an update of this belief based on additional information about this event. For example, assume a scenario in which the receiver of an image expects any of three digits (here: 0, 3, and 4) to arrive with equal probability. If the received image is highly corrupted, such that only a few of the received pixels are known to be correct, what can the receiver infer about the transmitted image? In this section, we let PSN networks provide the answer.

More specifically, we chose three images of digits 0, 3, and 4 from the MNIST dataset[54], rescaled to 12 × 12, and with brightness rounded to zero or unity. These images were mapped to a fully connected network of 144 PSNs, with one pixel assigned to each neuron. We then proceed to train the PSN network to store the prepared images, meaning the network will have three most likely states that form the images of the digits, with each state being equally probable. For this, we first train an equivalent BM using wake-sleep (see Sec. "*Probabilistic inference training*" in the Methods) and map the resulting parameters to the PSN network.

First, we assess the mixing capability of the network by observing its dreaming phase (corresponding to the sleep phase during training). Without external input, the PSN network switches randomly between states, forming the three images with approximately equal probability, which represents our prior belief about the possible images. Figure 7a shows a two-dimensional projection of PSN samples onto vectors corresponding to the images (see Sec. "*Probabilistic inference visualization*" in the Methods). We found that the result is close to that of the BM, and in most cases, the network switches between the numbers every few samples (Fig. 7b), indicating good mixing between the states.

Next, we assess the inference capability of the network in a pattern completion scenario. By applying additional bias to a few neurons that are only active for two out of the three patterns, we provide informative, but limited input to the network. We chose five pixels that are black for 0 and 3, but not 4 (Fig. 7c), and applied an additional positive bias to the corresponding neurons. The biases of other neurons are unchanged, i.e., no information is given. Such an input is ambiguous with respect. 0 and 3, but incompatible with 4. As a result, the PSN network only generates complete images of 0 and 3, and randomly switches between them.

## Learning from data

Sampling from arbitrary distributions implies the ability to sample from distributions dictated by real-world data. Here, we learn a generative model of handwritten digits based on the MNIST dataset[54]. Following ref. 20, we round the brightness of each pixel to the minimum or maximum.

To work with this dataset, we use a hierarchical sampling network: an RBM with three layers: visible, hidden, and label (Fig. 8a). The brightness of pixels is mapped to the activity of a neuron in the visible layer. The activity in the label layer shows which digit is represented in the visible layer at the current point in time. For example, if visible neurons form an image of 0, only the label neuron corresponding to a 0 will spike.

In this section, we consider three tasks: completion, guided dreaming and classification. For completion, visible neurons are clamped to the brightness of corresponding pixels except for a bottom-right quadrant, which is assumed obscured and remains free alongside other neurons. The task is for the free visible neurons to

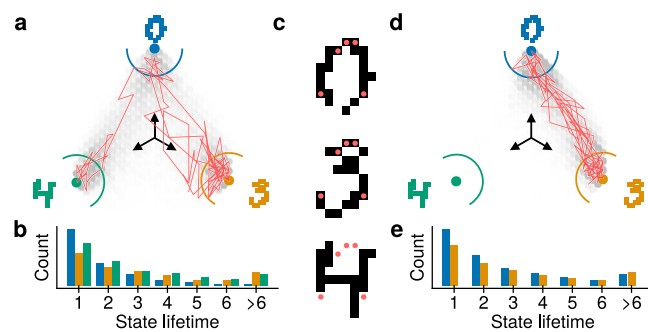

**Fig. 7 | Bayesian inference with a network of PSNs. a** Two-dimensional projection of network states (see Sec. "*Probabilistic inference visualization*" in the Methods) after sampling from the prior trained to store images of digits 0, 3 and 4. Colored dots show projections of the images. More samples are shown with darker dots; those inside the half-circles are considered to be close to a corresponding digit. The red line shows a trajectory of 100 consecutive samples. **b** Histogram of time in samples spent inside the half-circles in (a) (colors match). **c** Input for Bayesian inference. The five red markers show the pixels where positive bias is applied. Such an input is ambiguous w.r.t. 0 and 3, but incompatible with 4. **d**, **e** Same as (a, b) when provided the input shown in (c).

complete the obscured quadrant. For guided dreaming, one label neuron is clamped, and the visible neurons are expected to form an image of a corresponding digit. For classification, visible neurons are clamped to the brightness of corresponding pixels, and the most active label neuron on average is taken as the answer (winner-takes-all).

For these tasks, we use the BM parameters from ref. 55 (Fig. 8b,c), as they were already optimized for LIFS networks and are therefore expected to be favorable for an implementation with PSNs. This network is composed of 1194 neurons: 784 in the visible, 400 in the hidden, and 10 in the label layers, respectively. Its implementation with PSNs thus represents a scaling test for our general approach.

The simulation of PSNs during the tasks starts with a burn-in phase, where no input is provided. Then, the inputs are provided sequentially, with each subsequent input following immediately after the previous one.

For pattern completion with PSNs, 20 samples were drawn for each image. This is a difficult inference task, as the network should not simply produce an average image, but instead needs to adapt to the style of each individual input sample. Figure 8d shows the results for a few images from the MNIST testing dataset. We found that in most cases, the obscured parts were completed to a large degree of accuracy.

For guided dreaming, label neurons were clamped for $200\tau$. To enforce top-down control (from labels to pixels), we strengthened the weights between the hidden and label layers by a factor of 2, while marginally reducing the others by 10% (see Discussion). In Fig. 8e, we show how the PSN network can thereby be used for generating images from all learned classes.

While BMs are designed primarily as generative networks, especially when trained purely with contrastive Hebbian methods and without dedicated backpropagation-based fine-tuning, input classification can still be viewed as a form of Bayesian inference; thus, it is instructive to compare classification performance between the original BM and its PSN implementation.

Simulating the processing of 10,000 images of the testing MNIST dataset with a PSN network of this size is computationally intensive; we therefore drew samples until the saturation of the classification convergence curve (Fig. 8f). In this case, we drew 50 samples for each image, yielding an average classification accuracy of 87.0%. The RBM is much less demanding; we could thus draw 500 samples, obtaining 87.8% accuracy. Their confusion matrices are compared in Fig. 8g; we

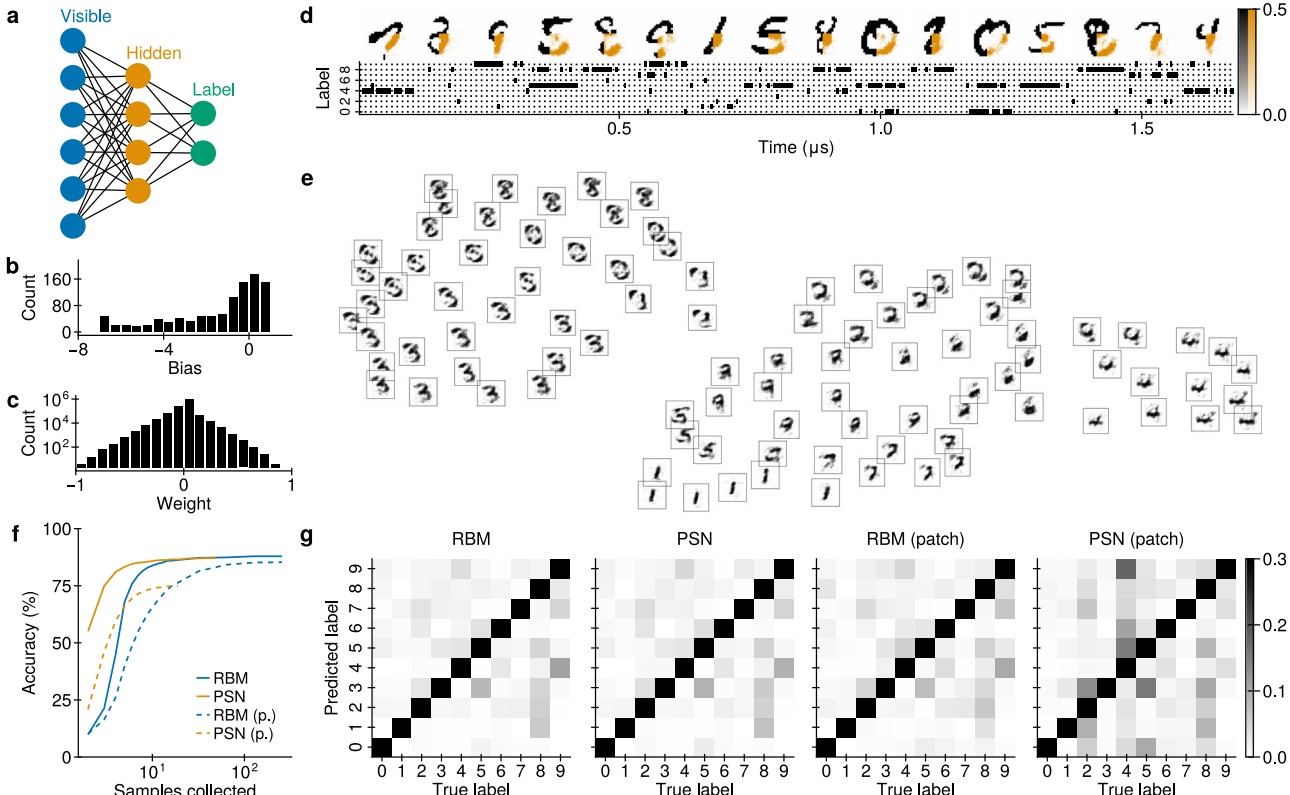

**Fig. 8 | Network of spiking nanolasers applied on the MNIST dataset. a** Scheme of an hierarchical sampling network. Each line represents a symmetric connection. **b**, **c** Histograms of Boltzmann machine parameters. Strongly negative biases (−30) were omitted from the figures. **d** Time trace of spiking nanolaser-based MNIST completion with a bottom-right quarter patch occlusion. Restored pixels are orange. **e** Guided dreaming. Each image corresponds to a separate dream. For each dream, the activity of neurons in the visible layer was averaged over the last 20τ of each dream. The image positions correspond to projection in two dimensions using t-SNE (see Sec. "*t-Stochastic Neighbor Embedding*" in the Methods). **f** Convergence of classification with an RBM and a corresponding PSN network with complete images (solid lines) and with the bottom-right quarter patch occluded (dashed lines). **g** Confusion matrices for classification with complete images (left pair) and the bottom-right quarter patch occluded (right pair).

note their similarity, as well as the only marginal performance loss caused by the exchange of substrates.

Interestingly, during the completion task, the networks perform classification as well. However, the accuracy of the PSN network is reduced to 75.0%, compared to 85.6% for the RBM (Fig. 8f); we expect this to be mitigated by further fine-tuning or direct in-situ training of the PSN network (see Discussion). However, in both cases, we observe that the PSN network only needs a few samples to converge to a solution, which is faster than the RBM and can prove beneficial in time-constrained scenarios.

## Discussion

In this work, we have demonstrated the feasibility of spike-based sampling using networks of photonic spiking neurons. We have rigorously derived an analogy between the gain dynamics of a two-section semiconductor laser and the membrane dynamics of biological neurons, both above and below the spiking threshold. Using the resulting translation rules, we have mapped the learned parameters of Boltzmann machines to the corresponding quantities in nanolaser networks and have demonstrated accurate sampling across varied tasks of different scale and complexity.

While the analogy between PSNs, ideal LIFS neurons, and ideal BM neurons represents a good approximation, we take note of two explicit differences. First, the refractoriness of LIFS neurons is absolute, i.e., a new spike cannot be emitted under any circumstances until the refractory period has passed. In PSNs, however, refractoriness is a result of a steep drop in the gain, and a strong enough excitation can result in premature spiking, i.e., the refractoriness of PSNs is relative.

This form of refractoriness is indeed more akin to the one found in biological neurons[40]. Second, the PSP shape in PSN networks is close to an alpha function. This represents a deviation from the interaction kernels in BMs, which are rectangular. Nevertheless, neither of these properties is significantly detrimental to the ultimate network sampling accuracy. This is in line with observations from refs. [11], [13], which also explicitly address the issues of relative refractoriness and PSP shape. We expect the accuracy to further improve when parameter translation is replaced by in-situ training of PSN hardware.

The use of photonic timescales fosters a large improvement of sampling speeds compared to biological or electronic timescales. Even for highly accelerated neuromorphic systems such as BrainScaleS-1[20] and BrainScaleS-2[10], convergence speeds for sampling from small Boltzmann distributions over 5 random variables amount to ca. 10 seconds. With PSNs, these convergence times drop by over 4 orders of magnitude to ca. $10^2$ microseconds. This acceleration factor would directly translate to a corresponding decrease in times-to-solution for any neuromorphic applications of spike-based sampling. Beyond the examples of Bayesian inference discussed here, these include tasks as diverse as stochastic constrained optimization[56] or quantum tomography[21,22].

In this work, we have considered photonic neuronal samplers ranging in size from a few PSNs to more than a thousand. While the implementation of individual components has seen recent experimental validation, the large-scale implementation of PSN networks in integrated photonics faces several challenges, which we discuss below.

We have assumed photonic crystal nanolasers as PSNs. Their small footprint promises high-density integration, but power dissipation then becomes an important issue. It is therefore necessary to reduce

the amount of power required for PSN operation. Although spiking nanolasers demonstrated so far are optically pumped, they can also be pumped electrically, and we expect they will require about 100 $\mu$A each[38]. For the PSN parameters used in this work, the required pumping is approximately 215 $\mu$A (see Sec. "*Nanolaser with quantum wells*" in the Methods). We estimate that for the MNIST tasks, 1194 nanolasers would require $3 \times 3$ mm$^2$ of chip space and, excluding electrical losses and controller power consumption, up to about 2 W: 250 mW for pumping and the rest for amplification of spikes (see Sec. "*MNIST power consumption*" in the Methods). With a sampling rate of approximately 0.09 GHz, such a network of PSNs is equivalent to an RBM running at 116 TFLOPS and 58 TFLOP J$^{-1}$ or 17 fJ FLOP$^{-1}$.

The photonic crystal nanolasers are complex optical structures that require a dedicated fabrication process. A recently demonstrated technique of micro transfer printing, where each cavity is transferred from a wafer on the chip, has been shown to be scalable[57].

The implementation of dense programmable optical interconnects is one of the most challenging and actively pursued goals. In this work, the largest interconnect matrix considered was $784 \times 400$. Establishing a fully connected network is, however, a challenge per se. One option is free space optics with programmable arrays of micro-mirrors and spatial phase modulators[58,59]; in this setting, a large number of channels can be controlled independently with a manageable level of losses. Spiking nanolasers such as those reported in[39] could be modified for free space emission using compact grating couplers or free-form optics with very low losses (-1 dB). On the other hand, micro-electromechanical systems (MEMS) offer ultra-low-loss switches connecting up to 240 ports[60]. We note that the MEMS technology implemented there offers digital switching, which is unsuitable for our purposes. However, a continuous control of the weights is possible by using MEMS-actuated directional couplers instead, with similar energy and loss specifications, as is the case for photonic matrices[24,25], which feature 64 channels and are fully reconfigurable and require low power. Finally, spectral multiplexing is possible owing to the incoherent nature of the interconnections, which could drastically increase the number of connections, as existing multiplexing/demultiplexing technology based on weighting filter banks handles several spectral channels[51,61]. Moreover, quantum computing and photonic accelerators are supporting the development of increasingly larger reconfigurable photonic matrices usable in the context of this work. Furthermore, three-dimensional interconnects[62] are also promising for more extreme cases and multi-chip systems.

Coherent coupling between lasers could happen due to residual (unwanted) reflection in the coupling matrix. Yet, we believe this effect could be made negligible as nanolasers can be individually designed to have a predetermined large spread of the emitted wavelengths[38]. The emitted frequencies could cover the full C-band (-10 THz), while the spectral width of the emitted spikes is well below 100 GHz. This way, the linear summation of the spikes in the photodiodes can also be ensured.

## Methods
### Nanolaser with quantum wells
The rate Eqn. (4) Describe the interaction of light with quantum dots[45,46]. These semiconductor nanostructures, akin to artificial atoms, localize carriers within a few nanometers and therefore are well modeled with a two-level electronic system. However, this work builds on the experimental results reported in ref. 39, where the gain material consists of quantum wells. There, unlike in quantum dots, the gain depends nonlinearly on the density of carriers in the conduction band (i.e. the population of excited dipoles in our model divided by the volume of the section). This nonlinear dependence is approximated by a piecewise linear function. The slope is larger at lower carrier density, with a ratio $\chi_g$. To ensure that $n_e \leq n_0$, we corrected the pumping term in Eq. (4) by replacing $n_e \rightarrow 2n_e/(\chi_g + 1)$.

The pump rate at threshold $\gamma_p$ is computed from Eq. (4) by assuming a steady state without noise, $G = 0$ and $S \approx 0$, which leads to $\gamma_p^{thr} = \gamma_t n_e^{thr}/[n_0 - 2n_e^{thr}/(\chi_g + 1)]$, where $n_e^{thr} = (\gamma + \gamma_{ra}n_{0,a} + \gamma_r n_0)/2\gamma_r$. Substituting values from Supplementary Table 1, we find $n_e^{thr} \approx 1.47 \times 10^6$ and $\gamma_p^{thr} \approx 1.31 \times 10^9$ that corresponds to a current of $\gamma_p^{thr} n_0 e \approx 215 \mu$A, where $e$ is the elementary charge.

### Nanolaser noise model
The stochastic Eq. (4) are composed of deterministic (drift) and stochastic (diffusion) parts that can be represented in matrix form:

$$du = \mu(u,t)dt + \sigma(u,t)dW_t,\tag{13}$$

where $u = [S, n_e, n_a]^T$, and $W_t$ is the Wiener process. In this work, the diffusion term follows the approach in [ref. 47, A.13.1.2] based on the McCumber noise model[63] and is comprised of five Langevin forces corresponding to the following groups of processes:
1. electron-photon interaction in the gain section,
2. same, in the SA,
3. electronic processes inside the gain section,
4. intrinsic optical loss,
5. electronic processes inside the SA.

The Langevin forces are stochastic processes represented with Wiener processes with zero average and cross-correlation $2D_{ij}$; for $i = j$, the latter represents the autocorrelation or noise spectral density. For the groups of processes described above, we find

$$
\begin{aligned}
2D_{SS}^e &= \gamma_r n_0 S + \gamma_r n_e,\\
2D_{SS}^a &= \gamma_{r,a} n_{0,a} S + \gamma_{r,a} n_a,\\
2D_{ee}^o &= \gamma_p\left(n_0 - \frac{2n_e}{1+\chi_g}\right) + (\gamma_t - \gamma_r)n_e,\\
2D_{SS}^\gamma &= \gamma S,\\
2D_{aa}^o &= (\gamma_{t,a} - \gamma_{r,a})n_a,
\end{aligned}\tag{14}
$$

where $n_e$, $n_a$ and $S$ are the averaged (not stochastic) values. This way, the diffusion matrix becomes:

$$\sigma(u,t) =$$

$$
\begin{bmatrix}
\sqrt{2D_{SS}^e} & \sqrt{2D_{SS}^a} & & \sqrt{2D_{SS}^\gamma} & \\
-\sqrt{2D_{SS}^e} & & \sqrt{2D_{ee}^o} & & \\
& -\sqrt{2D_{SS}^a} & & & \sqrt{2D_{aa}^o}
\end{bmatrix}.\tag{15}
$$

The equations are integrated using the *SKSROCK* solver from the *DifferentialEquations.jl* library[64] in the Julia programming language[65].

### Simplified gain equation
Here, we derive the Eq. (6). Consider a network of PSNs. A derivative of the gain of a PSN is given in Eq. (4):

$$\dot{G} = 2\gamma_r \dot{n}_e + 2\gamma_{r,a}\dot{n}_a.\tag{16}$$

Assume the PSN is not currently emitting a spike, i.e. its gain does a random walk shown in Fig. 2c, in which case, $S \approx 0$. Then, substitute the derivatives of electron populations from Eq. (4):

$$
\begin{aligned}
\dot{G} &\approx -2\gamma_r\gamma_t n_e + 2\gamma_r(\gamma_p + \Delta\gamma_p(t))(n_0 - n_e)-\\
&\quad - 2\gamma_{r,a}\gamma_{t,a}n_a + 2\gamma_r F_e(t) + 2\gamma_{r,a}F_a(t) =\\
&= -2\gamma_r n_e(\gamma_t + \gamma_p) + 2\gamma_r\gamma_p n_0 - 2\gamma_{r,a}\gamma_{t,a}n_a +\\
&\quad + 2\gamma_r\Delta\gamma_p(t)(n_0 - n_e) + 2\gamma_r F_e(t) + 2\gamma_{r,a}F_a(t).
\end{aligned}\tag{17}
$$

where $\Delta\gamma_p(t)$ is given in Eq. (5). Then, replace $2\gamma_r n_e = G + \gamma - \gamma_{r,a}(2n_a - n_{0,a}) + \gamma_r n_0$:

$$\dot{G} \approx -(\gamma_t + \gamma_p)(G + \gamma - \gamma_{r,a}(2n_a - n_{0,a}) + \gamma_r n_0)- \\ + 2\gamma_r\Delta\gamma_p(t)(n_0 - n_e) + 2\gamma_r\gamma_p n_0 - 2\gamma_{r,a}\gamma_{t,a}n_a + \\ + 2\gamma_r F_e(t) + 2\gamma_{r,a}F_a(t). \tag{18}$$

Rearranging the terms we find

$$\dot{G} \approx -(\gamma_t + \gamma_p)G + 2\gamma_{r,a}(\gamma_t + \gamma_p - \gamma_{t,a})n_a- \\ - (\gamma_t + \gamma_p)(\gamma + \gamma_{r,a}n_{0,a} + \gamma_r n_0) + 2\gamma_r\gamma_p n_0- \\ + 2\gamma_r(n_0 - n_e)\Delta\gamma_p(t) + 2\gamma_r F_e(t) + 2\gamma_{r,a}F_a(t). \tag{19}$$

The second term on the first line is negligible compared to the first, as during the random walk $n_a \ll n_e$. The terms on the second line can be considered a drift term:

$$G_p = -\gamma - \gamma_{r,a}n_{0,a} - \gamma_r n_0 + 2\tau_G\gamma_r\gamma_p n_0, \tag{20}$$

where $\tau_G = 1/(\gamma_t + \gamma_p)$. This way, the dynamical equation becomes

$$dG \approx \left[ -(G - G_p)/\tau_G + 2\gamma_r(n_0 - n_e)\Delta\gamma_p(t) \right] dt + \sigma_G dW. \tag{21}$$

Here, the interaction term can be nonlinear as $n_e$ changes due to incoming spikes; we assume that such an interaction is weak. Moreover, during the walk, the gain section is close to transparency, i.e. $n_e \approx n_0/2$. Finally, we find

$$dG \approx \left[ -(G - G_p)/\tau_G + 2\gamma_r n_0\Delta\gamma_p(t) \right] dt + \sigma_G dW. \tag{22}$$

## Discrete nanolaser model

In this work, we assumed a continuous model, i.e., $S$, $n_e$, and $n_a$ are continuous variables, and stochastic processes are approximated by Langevin forces. However, the particles are discrete, and so are the processes. In a typical laser, the number of photons and electron-hole pairs in lasers is large, and such a model is a good approximation. However, for nanolaser,s this can be disputed, given their small volume and the operation regime close to the threshold. Moreover, in the model used here, noise can cause the variables to leave the physical range of values, i.e. $S$, and $n_e$ and $n_a$ can become negative, which can cause unwanted changes to the parameters of noise (see Eq. (14)). We force $S$ to be above a small positive number and $n_e$ and $n_a$ are always at least 0. This workaround may significantly change the stochastic behavior of the model. Therefore, we consider a discrete model based on refs. 46,66 which does not suffer from such issues.

We consider all processes separately – 10 total – as stochastic with rates: $\gamma_{r/r,a}Sn_{e/a}$ for stimulated emission in the gain section / SA, $\gamma_{r/r,a}S(n_{0/0,a} - n_{e/a})$ for photon absorption in the gain section / SA, $\gamma S$ for optical loss, $\gamma_{r/r,a}n_{e/a}$ for spontaneous emission in the gain section / SA, $\gamma_{t/t,a}n_{e/a}$ for nonradiative recombination and out-of-mode spontaneous emission in the gain section / SA, and $\gamma_p(n_0 - 2n_e/(\chi_g + 1))$ for pumping. When an event happens, a single particle is added or removed from the appropriate variables. For example, for stimulated emission in the SA, $S \to S + 1$ and $n_a \to n_a - 1$. Such a simulation is considerably more computationally demanding, but is rigorous and more accurate for this system.

The simulation was carried out using the *SSAStepper* solver from the *DifferentialEquations.jl* library[64]. Supplementary Fig. 1 shows a comparison between the models. We find that they give very similar results in the operating regime of interest.

## Contrastive divergence

We use contrastive divergence to train an RBM to sample from an arbitrary distribution $p^*(z)$. In this section, we provide a brief summary of the procedure; for a more detailed introduction, we refer to ref. 67.

We start with arbitrary weights $\mathbf{W}$, visible biases $\mathbf{b}_h$ and hidden biases $\mathbf{b}_h$. Then, we iterate over all possible states of $\mathbf{z}$ and set the target probabilities to $p^*(\mathbf{z})$. For each $\mathbf{z}^m$, we set the initial state of visible neurons $\mathbf{v}(0) = \mathbf{z}^m$. Following the RBM sampling procedure, we then sample the hidden neurons $\mathbf{h}(0)$ given $\mathbf{v}(0)$, followed by a resampling of the visible states $\mathbf{v}(1)$ given $\mathbf{h}(0)$, etc. This procedure is repeated $K$ times, resulting in a sequence of states $\mathbf{v}(0, ..., K)$ and $\mathbf{h}(0, ...,K)$. The final states $\mathbf{v}(K)$ and $\mathbf{h}(K)$ are then used for updating the parameters. In our case, $K$ was also gradually increased from 1 up to 20 as the sampling accuracy improved.

For each $\mathbf{z}^m$, the state-specific parameter updates are then calculated, but not yet applied. The weight updates are given by

$$\Delta\mathbf{W}^m = p^*(\mathbf{z}^m)(\mathbf{v}(0) \otimes \mathbf{h}(0) - \mathbf{v}(K) \otimes \mathbf{h}(K)), \tag{23}$$

where $\otimes$ is the outer product; for the bias updates, we have

$$\Delta\mathbf{b_v}^m = p^*(\mathbf{z}^m)(\mathbf{v}(0) - \mathbf{v}(K)), \\ \Delta\mathbf{b_h}^m = p^*(\mathbf{z}^m)(\mathbf{h}(0) - \mathbf{h}(K)). \tag{24}$$

After iterating through all $m$, the weights and biases are updated according to $\Delta\mathbf{W} \propto \sum_m \Delta\mathbf{W}^m$, $\Delta\mathbf{b}_v \propto \sum_m \Delta\mathbf{b}_v^m$, and $\Delta\mathbf{b}_h \propto \sum_m \Delta\mathbf{b}_h^m$.

## Probabilistic inference training

The approach used here follows[13]. A fully visible BM was trained to store three digits – 0, 3 and 4 – taken from the MNIST dataset and scaled down to $12 \times 12$. The intensity of each pixel, which ranged from zero to unity, was rounded to 0.05 or 0.95. This way, the intensity of the pixels defines the target statistics $\langle z_k \rangle^*$ and $\langle z_k z_j \rangle^*$ for the BM. We start with arbitrary weights $W_{kj}$ and biases $b_j$ and collect a sufficient number of samples to estimate $\langle z_k \rangle$ and $\langle z_k z_j \rangle$. Then, we refine the BM parameters using the following update rules: $\Delta b_k \propto \langle z_k \rangle^* - \langle z_k \rangle$ and $\Delta W_{kj} \propto \langle z_k z_j \rangle^* - \langle z_k z_j \rangle$. Then, sampling and refinement is repeated until a satisfactory result is achieved.

## Probabilistic inference visualization

Figure 7 a,d visualize high-dimensional states of the network using a star plot of their projections. The projection procedure follows[13] with a minor modification.

The three axes show the basis vectors $\mathbf{B}$ representing pixel intensities of the target images:

$$\langle z_k \rangle = (\mathbf{B}^0, \mathbf{B}^3, \mathbf{B}^4)^T. \tag{25}$$

The basis vectors are normalized: $||\mathbf{B}^i|| = \sqrt{\sum_j |B_j^i|^2} = 1$. After a simulation, the network states are discretized once per $\tau$ and are projected onto this basis:

$$\mathbf{z}^{034}(t_m) = \left( \mathbf{B}^0 z(t_m), \mathbf{B}^3 z(t_m), \mathbf{B}^4 z(t_m) \right)^T. \tag{26}$$

This projection is, in turn, projected onto the two-dimensional plane $\mathbf{z}^{proj}(t_m) = \mathbf{M}^{proj}\mathbf{z}^{034}(t_m)$, where

$$\mathbf{M}^{proj} = \begin{bmatrix} \sin(\varphi_B^0) & \sin(\varphi_B^3) & \sin(\varphi_B^4) \\ \cos(\varphi_B^0) & \cos(\varphi_B^3) & \cos(\varphi_B^4) \end{bmatrix} \left( \mathbf{B}^T\mathbf{B} \right)^{-1}, \tag{27}$$

where $\mathbf{B} = (\mathbf{B}^0, \mathbf{B}^3, \mathbf{B}^4)$ and $(\varphi_B^0, \varphi_B^3, \varphi_B^4) = (0, 2\pi/3, 4\pi/3)$. The $\left( \mathbf{B}^T\mathbf{B} \right)^{-1}$ term accounts for the non-orthogonality of the basis vectors and was not used in[13].

Due to a limited number of unique projection results, a standard scatter plot cannot represent clustering of a large number of projections in a single spot. Instead, we opt for a scatter plot with large near-transparent markers. This way, multiple overlaid projections result in a darker marker.

The proximity of the network states to the target states is represented by the distance between $z^{proj}(t_m)$ and $\langle z_k \rangle$. Judging by the clustering of dark markers around the projections of the target images, the network spends the majority of time around the corresponding states.

## MNIST power consumption

First, we estimate the power necessary to reach the operating point, which, in PSNs, is close to the threshold (see Fig. 4a). For the parameters considered here, it is $\gamma_p^{thr} \approx 215\ \mu A$ (see Sec. "*Nanolaser with quantum wells*" in the Methods). The current due to BM bias will change the electrical current by a few percent (see Fig. 4a), which we consider negligible. The typical bias voltage for III-V semiconductor lasers is 1 V, which results in approximately $215\ \mu W$ per PSN. For MNIST with 1194 nanolasers, 250 mW would be required, although roughly 20% of PSNs never spike and require no pumping.

Next, we estimate the power necessary for interaction between PSNs. Assume the $j$-th PSN has emitted a spike that is then routed to the $k$-th PSN. According to Eq. (4), $\Delta n_{e,k} = \int \Delta\gamma_{p,k}(t)(n_0 - n_e)dt$. Assume the $k$-th PSN is undergoing a random walk, during which $n_{e,k} \approx n_0/2$, and with Eq. (5), we find

$$\Delta n_{e,k} \approx \frac{n_0}{2} \iint \kappa_{kj} \mathrm{LPF}(t - t^*) S_j(t^*) dt^* dt. \tag{28}$$

Since the low-pass filter preserves energy, the two integrals become the total photon count of a spike; overall, $|\kappa_{kj}|n_0/2$ of the photons are absorbed.

Now, assume the spike is subject to losses in the transmission chain, and the $k$-th PSN only receives a fraction $\zeta$ of the expected photons; thus, $\zeta^{-1}$ times more photons must be sent to compensate. Summing over all PSNs connected to the $k$-th, we find that $\zeta^{-1}\sum_j |\kappa_{kj}|n_0/2$ of the spike photons must be sent to the coupler. If this value exceeds unity, more photons are required than the spike contains, and amplification is therefore required. With the parameters considered in this work, the translation rule for weights is $\kappa_{kj} \approx (0.2/n_0)W_{kj}$. Then, the $k$-th neuron must send $0.1 \cdot \zeta^{-1}\sum_j |W_{kj}|$ of the spike power, and the required amplification multiplier is

$$\max\left[1, 0.1 \cdot \zeta^{-1}\left(\sum_j |W_{kj}|\right) - 1\right]. \tag{29}$$

The weighting scheme is expected to be the dominant source of optical losses. Here, we follow the conclusion in the previously mentioned work on a 240 × 240 switch[60]. There, losses are dominated by waveguide crossings: 0.016 dB for each. For MNIST, a spike traveling between the visible and the hidden layers would pass over 784 + 400 − 1 crossings, which corresponds to 18.9 dB loss, and, similarly, 6.5 dB for the hidden and label layers. This loss is expected to be significantly reduced by the use of multilayer bus waveguides. As a result, propagation losses become dominant, but the use of shallow-etched silicon rib waveguides allows losses down to 0.1 dB cm$^{-1}$. For the visible-hidden coupler, it corresponds to 0.9 dB of loss ( = 0.1dB × 75 $\mu$m cell$^{-1}$ × 1184 cells) and 0.3 dB for the hidden-label coupler. Still, with the technology in[60] and such waveguides, for the visible-hidden coupler we find $\zeta \approx -19.8$ dB and for the hidden-label coupler $\zeta \approx -6.8$ dB. Then, using Eq. (29). We find the average amplification

- 24.2 dB from the visible to the hidden layer,
- 27.1 dB same, in reverse,
- 0.002 dB from the hidden to the label layer,

- 14.2 dB same, in reverse.

To get amplification in power units, we must first find the power emitted by a PSN. In Fig. 1b, we find that $S(t)$ can be approximated with a Gaussian with FWHM ≈ 30 ps. The peak photon count in the cavity is $S_{max} \approx 8e4$. Assuming that intrinsic optical losses are negligible compared to losses due to waveguide coupling, photons escape to the waveguide at a rate $\gamma S(t)$. Considering that photons escape in both directions of the waveguide (see Fig. 1a), the number of useful photons is halved, $\gamma S_{max}$ FWHM $\sqrt{\pi}/2 \approx 4.2e5$ which at C-band (1550 nm) corresponds to 54 fJ. The emission power depends on emission frequency, which can vary significantly depending on the dynamics of the entire network. For simplicity, assume that all PSNs operate with zero membrane potential, i.e., $p(z = 1|b = 0) = 0.5$. Then, the average power emitted by a PSN is 54fJ × 0.5 × 0.1GHz ≈ 2.7 $\mu$W. Therefore, amplification of the signals from the visible layer to the hidden layer would require 2.7 $\mu$W × 784 × 24.2dB ≈ 560mW. Following this logic, we find:

- 560 mW from the visible to the hidden layer,
- 1100 mW same, in reverse,
- 2 mW from the hidden to the label layer,
- 55 mW same, in reverse.

Therefore, for MNIST classification, where the visible layer is clamped, amplification of spikes coming from the hidden to the visible layer is unnecessary, and 560 mW + 2mW + 55 mW = 617 mW is required for amplification in total. In contrast, during guided dreaming, the label layer is clamped and 560 mW + 1100 mW + 55 mW = 1715 mW are required.

For photonic accelerators, floating point operations (FLOP) per second (FLOPS) are often provided as a figure of merit. For the network of PSNs, it is appropriate to find how many FLOPs is required by the BM it replicates. For simplicity, the hierarchical sampling network used here can be considered an RBM, where visible and label neurons are in one layer, and hidden neurons are in the other layer. The neuron activation function is a sigmoid $\sigma(x) = \exp(x)/(1 + \exp(x))$. Assuming that exp can be computed in 1 FLOP, computing the sigmoid costs 3 FLOPs. A neuron connection is a multiplication and requires 1 FLOP.

During MNIST classification, 65% of neurons are clamped, and the estimation of FLOPS will not represent the true capability of the PSN network. Instead, consider drawing one sample during the guided dreaming task. First, the activations of the visible and the label neurons are weighted by a 400 × (784 + 10) matrix, which requires 2 × 400 × 794 FLOPs. Then, the activations of the hidden neurons are computed by adding 400 biases and spending 3 × 400 FLOPs for the sigmoid. The same logic holds for the backward step. Therefore, one sample requires 2 × 400 × 794 + (1 + 3) × 400 + 2 × 794 × 400 + (1 + 3) × 794 = 1275176 FLOPs. Considering the network of PSNs draws one sample per $\tau$ = 11 ns, it performs an equivalent of 1275176FLOP/11ns ≈ 116 TFLOPS during this task. With the previously found approximate power draw of 2 W, we also find the energy efficiency of 58 TFLOP J$^{-1}$ or 17 fJ FLOP$^{-1}$.

To conclude, the power consumption is composed of PSN pumping and spike amplification. For MNIST tasks, the former is 250 mW for the parameters used in this work. The latter, assuming crossbar technology used in ref. 60, varies between 600 mW and 2 W depending on the task. The performance is estimated to reach 116 TFLOPS with the energy efficiency of 58 TFLOP J$^{-1}$ or 17 fJ FLOP$^{-1}$.

## t-Stochastic Neighbor Embedding

t-Stochastic Neighbor Embedding (t-SNE) is a technique for dimensionality reduction of highly-dimensional datasets[68]. In particular, it performs a nonlinear projection of data on a two-dimensional plane, where the proximity of the projections represents the similarity between the features of the data points.

During guided dreaming, the PSN network generates images of digits. Due to stochasticity, the generated images are not identical, but still have similar features, which are used by t-SNE to cluster the images for better visualization.

## Simulation code details

In order to apply a BM bias $b$ to a neuron in the PSN network, instead of Eq. (10) We translate the bias to PSN pumping directly using

$$\gamma_p = \gamma_{p,0} + \partial_u \gamma_p b, \tag{30}$$

which is equivalent as $G_p$ and $\gamma_p$ are linearly dependent. This is a more practical approach that allows using Fig. 4a directly.

All tasks considered in this work follow the same pattern (see also Supplementary Fig. 2):

1. train a BM to solve a problem,
2. translate BM parameters to PSN parameters,
3. simulate the PSN network,
4. discretize the PSN output,
5. process the result as if the output is from a BM.

Additional parameters related to tasks and tuning of PSNs are given in Supplementary Table 2.

## Data availability

The parameters of the RBM used for sampling from an arbitrary distribution, the distribution itself and the BM used for probabilistic inference generated in this study have been deposited in the Zenodo database under accession code https://doi.org/10.5281/zenodo.17450281. The RBM used for the MNIST tasks is in the supporting dataset of ref. 55. The MNIST dataset is publicly available[54].

## Code availability

We are open to receiving individual requests for direct access to the code. We will meet any such request with our full support, but our institutional regulations require us to obtain case-by-case approval from the management of Thales.

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

## Acknowledgements

I.K.B. acknowledges support by ANR, project INFERNO, partial support by the European Union through the Marie Sklodowska-Curie Innovative Training Networks action POST-DIGITAL, project number 860360, and partial support from EU Commission Chips Joint Undertaking, project number 101194363 (NEHIL). M.A.P. gratefully acknowledges support from the Horizon Europe grant agreement 101147319 (EBRAINS 2.0) and the continuing support from the Manfred Stärk Foundation for the NeuroTMA Lab.

## Author contributions

I.K.B. performed simulations, analyzed the data, designed the figures, and wrote the initial draft of the paper. A.d.R. and M.A.P. conceived and supervised the project on photonic and neural network aspects, respectively. A.d.R. developed the model for the spiking nanolaser. M.A.P. designed the experiments and methods for data analysis and visualization. All authors contributed to the article and approved the submitted version.

## Competing interests

The authors declare no competing interests.
