## [Transparent Peer Review file · Nature Communications]

Ultrafast neural sampling with spiking nanolasers

Corresponding Author: Dr Ivan Boikov

Version 0:

Reviewer comments:

Reviewer #1

(Remarks to the Author)

The manuscript proposes and analyzes the concept of spiking nanolasers, i.e. semiconductor lasers that emit ultrafast optical pulses, and demonstrates a model for computing based on sampling-based neural computation. The core idea is to use small photonic crystal nanolasers, each behaving as a stochastic spiking neuron, to perform Bayesian inference through sampling from learned probability distributions (e.g., Boltzmann machines). The authors provide a theoretical treatment comparing these nanolaser-based neurons to conventional leaky integrate-and-fire sampling (LIFS) neurons, show how to map typical Boltzmann machine parameters onto spiking nanolasers, and then illustrate the approach on generative tasks and pattern completion (e.g., small-scale MNIST).

Key contributions include:

- i) A derivation that links nanolaser gain dynamics and standard stochastic neuron models.
- ii) Numerical experiments suggesting significant speed and potential power-efficiency gains over purely electronic spiking systems.
- iii) Demonstrations of sampling accuracy for small toy problems (MNIST-like tasks).

Overall, the paper provides an interesting proposal for ultrafast, optically based neuromorphic systems. The results are timely, given the growing interest in photonic computing for machine learning, and the manuscript's focus on sampling-based models (rather than purely feedforward neural nets) highlights unique advantages of spiking neural substrates.

2. Significance and Relation to Established Literature

Optical spiking neurons could become a promising direction in neuromorphic engineering due to their potential for high-speed operation and parallel data transmission. Compared with other photonic neuromorphic proposals, the authors' emphasis on sampling, as opposed to deterministic classification, addresses an important niche. This aspect could be further empathized in the introduction.

The paper references prior demonstrations of spiking semiconductor lasers and places the present approach in context with existing neuromorphic devices like BrainScaleS and other hardware for spike-based sampling.

Provided these speed and energy claims hold up in experimental settings, this work may have significant impact on the design of specialized photonic or hybrid photonic-electronic chips for Bayesian inference, optimization, and other tasks that benefit from fast random sampling. It builds upon established knowledge of Boltzmann machines and spiking neural networks but extends it into the realm of ultrafast photonic hardware.

3. Support for Conclusions and Claims

Spiking Mechanism & Neural Analogy: The theoretical equations linking the gain parameter of the laser to a membrane-potential-like stochastic variable appear consistent with known rate equations for semiconductor lasers. The authors' simulations further support that the excitability and refractoriness can mirror spiking neuron behavior.

Sampling Accuracy: Simulation results show that mapped Boltzmann parameters lead to correct sampling distributions in small networks, as well as decent performance on pattern completion for MNIST-like inputs. These results support the claim that photonic spiking neurons can implement sampling-based inference.

Energy and Performance Estimates: The authors provide speed estimates (on the order of gigahertz spiking rates), as well as approximate energy calculations that suggest orders-of-magnitude gains over purely electronic solutions. While these numbers are plausible, an experimental prototype or a more explicit breakdown of power budgets (e.g., how pumping, waveguide coupling, photodiode readouts, and any amplification circuitry scale with network size) would further validate the claims.

Additional Evidence: The paper convincingly motivates the theoretical viability of photonic sampling. However, real-world

constraints, fabrication tolerances, optical losses, device mismatch, and large-scale routing complexities, might introduce deviations from idealized modeling. Some more in-depth discussion of those potential challenges, or references to relevant integration technologies, would strengthen the claims.

Please address in the discussions points b, c, d and how these could be overcome and or limit the current work.

Potential Flaws and points that require revision

a) Energy Estimation Details: The manuscript's power estimates for large-scale networks could benefit from more specificity on amplifier or detector overheads, along with tolerance analysis of waveguide losses. As it stands, the rough estimates seem encouraging but may need tighter bounds or a more explicit breakdown.

b) Definition of Abbreviations: There are multiple instances where abbreviations appear before being defined. A comprehensive glossary or ensuring every acronym is defined upon first appearance would significantly improve clarity. (eg, SPS, PNS page 2 etc..)

c) Methodological Completeness: While the theory and simulation approach are well described, readers from the photonic hardware community may want more details on device geometry or coupling schemes. Additional clarity about how well the authors expect the assumptions (e.g., linear summation of spikes, negligible inter-laser coherence) to hold in practice would help reproducibility.

5. Methodology and Reproducibility

Soundness: The approach, using the known rate equations for semiconductor lasers augmented with noise terms, is standard and appears methodologically sound. Mapping Boltzmann parameters onto laser parameters follows from typical sampling-based spiking neuron theory.

Reproducibility: The methods section largely includes the relevant equations and standard parameter sets, enabling future researchers to replicate the simulations. Including a fully annotated list of parameters for each experiment (e.g., for the MNIST demonstration) and clarifying how the code is structured would enhance reproducibility.

In summary, the authors present a theoretically sound proposal. The authors make a strong case for how such ultrafast optical spiking neurons could perform sampling-based inference with notable improvements in speed and potentially power efficiency. Despite some points that would benefit from more details, particularly around practical power budgets and large-scale integration, the paper is overall a valuable contribution.

Recommendation: I recommend revisions to polish the manuscript, clarify energy estimates, ensure consistent definitions of abbreviations, and possibly strengthen the discussion around potential integration and scaling challenges.

Reviewer #2

(Remarks to the Author)

This work reports on the mapping from a photonic spiking neural network (PSN) to a Boltzmann machine (BM), and numerically demonstrates efficient computational tasks based on Bayesian inference. Among the myriad of today's optical computing approaches, this paper addresses the neuromorphic spiking response of semiconductor lasers with saturable absorbers, as a consequence of excitability. While excitability in semiconductor lasers is quite an old topic, intensively investigated in the past two decades, the advent of integrated photonic devices on the one hand, and the groundbreaking advancements in AI on the other hand, have boosted the research in this domain. In this respect the present manuscript represents quite a significant advancement. In particular, the small footprint and low consumption of photonic crystal nanolasers are claimed to offer a clear advantage over in-silico substrates, both in terms of speed and energy budget. Nevertheless, there are a few important issues that prevent me to positively recommend this work for publication in Nat. Comm. (points 1-4 below).

1) Broad audience and general interest. This manuscript is difficult to be understood by a non-specialist. It assumes a reader's background on Bayesian statistics and Boltzmann machines, which is far to be the case even in the community of optical neuromorphic computing. It is then of primary importance to introduce the main concepts of Bayesian inference, Boltzmann machines and sampling (both non-spiking and spiking), with a basic example (prior to spiking models) which would help the reader to a large extent. Otherwise, we might consider this work to be better suited to a more specialized journal (JCP or CPC). In my opinion, introducing these concepts is substantially more important than the details of the mapping ("Membrane potential of spiking nanolasers"): most of this subsection could be moved to a supplementary material, retaining only the fundamental result in the main text.

2) Overall clarity. Many points remain obscure to me (many of them in connection to point 1). For instance: it is not clear how the training is performed, i.e. what parameters (weights?) are tuned, and how. Another example is the 2D representations of network states (Fig. 7.a and 8.e): the vast majority of readers ignore what T-SNE is, so a hint to interpret the representation axes is needed. Furthermore, sentences like "...the PSN network correctly samples from the prior" (l. 410) or "...by sampling from conditional probability distributions" (l. 345), or "the probability of each state sampled from the inverse..." (l. 378) are very technical and specific from Bayesian statistics, so they need further clarification here. How PSP is implemented is not clear neither.

3) The optical spiking advantage. It is claimed that the spiking dynamics is advantageous both in terms of energy saving and

robustness to noise. While I acknowledge this as general statement, which may hold true in neuroscience, it is not clear to what extent it applies to the particular photonic system studied here. Specifically:

a. Power. CW laser-threshold powers are even smaller than those of pulsed devices.

b. Bandwidth. Frequency multiplexing is not clear in the context of the proposed photonic architecture, and should be commented.

c. Losses. Interconnect matrix is based on waveguide crossbar arrays. Then the scaling of crossing losses is a potential issue in large networks, which is not discussed.

4) The context of excitability in lasers. The references in the Introduction do not fairly reflect the historical developments. The first cited works on excitability (Refs. [23-25], Wunsch et al., Romeira et al., Prucnal et al.) do not accurately retrace the seminal studies on optical excitability. The first report on excitability was by F. Plaza et al. (EPL 38, 85, 1997) in a CO₂ laser with SA, followed by experiments in semiconductor lasers (M. Giudici et al., Phys. Rev. E 55, 6414, 1997) and Nd-YAG lasers with SA (Larotonda et al., PRA 65, 033812, 2002). In the context of photonic crystals, the first demonstration in a nanocavity was reported in Brunstein et al, PRA 85, 031803 (2012).

Minor comments:

i. Abstract. Saying that speed is an advantage of optics w.r.t electronics is not true in general, as an electronic signal also propagates at the speed of light in metallic wires. In addition, in the abstract (l. 5), what do authors mean by "conventional semiconductor devices"?

ii. Possible confusion in the notation: β is used as spont. emission factor of nanolasers, and as a parameter for probability distributions.

iii. Absolute/relative refractoriness are assumed to be well-known concepts, yet they're most likely ignored by the general reader.

iv. Line 264 should read: "Second, changes of n_e due to incoming spikes need to be small..."

Reviewer #3

(Remarks to the Author)

The manuscript demonstrates the feasibility of general spike-based sampling using a network of spiking lasers and discuss several examples of this sampling operation for the realization of complex machine learning tasks. The work is very complete, very well written and should be interesting to a large community ranging from machine learning specialists to photonics experts. The results include foundational points such as how to map the spiking probability to BM weights and very advanced tasks such as "guided dreaming". For the completeness of the work, its novelty, informative content and general quality, I warmly recommend publication of the manuscript.

However, a few points deserve clarification in my opinion:

The coupling between nodes is considered to take place via embedded photodetectors, aiming at incoherent transmission of spikes. Is coupling through saturation of the absorber section impossible or impractical?

The model uses variable S to describe a photon number, which must therefore be positive. What prevents S from becoming negative in presence of the stochastic term F ? This is particularly relevant considering that most of the time the system is not emitting a spike thus in absence of noise $S=0$ (figure 1b).

Figure 2b shows time traces of the normalized gain, which is, as expected mostly slightly below 0. However, at the onset of spikes, the gain depletion is anticipated by a very large increase of the gain (particularly visible for instance on spikes 3,4,5), which takes place at a time scale which is comparable to the following depletion (which is mediated by the photon number growth). What is the origin of this very sharp increase? Could it be related to the previous point?

The "wake sleep" algorithm *must* be briefly described, with a reference for more information, if the work is to be readable also to non-experts in stochastic neural networks training.

Points to be considered:

The "optical probabilistic inference" and "learning from data" sections seemed to me as "logical" specific applications of the section "optical sampling from arbitrary distribution". If it is the case, I fear these sections do not really help the reader. I understand they constitute "bright" points and their implementation is probably worth a description but if they are really direct consequences of "optical sampling from arbitrary distribution" the reader would benefit from being informed of that very clearly.

Minor points:

figure 1a: is a spike really shorter than the laser resonator as suggested by fig 1a? If yes, a PDE system would really be required to describe the physics. If not, then a single mode model as used here is fine but then figure 1a is a bit misleading.

"alpha shaped" is used here and there in the manuscript but not defined

The connection between BMs and spiking networks cites 11-13, I believe Hinton, G. E., & Brown, A. (1999). Spiking

boltzmann machines. Advances in neural information processing systems, 12. should be credited.

Version 1:

Reviewer comments:

Reviewer #1

(Remarks to the Author)

I thank the authors for thoroughly addressing all of the substantive points raised: they have strengthened the introduction by emphasizing the sampling niche with new citations; provided a detailed "MNIST power consumption" section to clarify energy budgets; discussed fabrication tolerances and routing challenges by covering noise tolerance, photodiode resilience, and options such as free-space/MEMS/spectral multiplexing; cleaned up infrequent acronyms, fixed hyperlink definitions, and added a "Simulation code details" section with annotated parameters to improve methodological clarity and reproducibility.

A single follow-up question remain: do the authors intend to publicly release their simulation code, and how should readers interpret the statement "Simulation code is available upon a reasonable request"? Could the authors clarify under what conditions and through what process one might request and obtain the code?

Reviewer #2

(Remarks to the Author)

Thank you for having modified the manuscript according to my comments and questions. Most of the "obscure points" have now been clarified in the text. Still there are a couple of issues that need to be addressed:

1) The "optical spiking advantage". The authors' response to my point 3a) is: "we do not believe the argument is applicable in the context of this work: CW lasers would not be suitable for stochastic neural networks due to a lack of prominent stochasticity.". This is not true for a nanolaser: close enough to laser threshold, the amplified spontaneous emission noise dominates the dynamics. Therefore, the question remains open: are spikes in optical NNs advantageous enough –as they are in the brain–, justifying its implementation over other (stochastic) approaches in a photonic context? I think a short discussion on this point is needed.

2) The authors have revised their use of the word "speed", and now focus on "bandwidth" and/or "latency". Unfortunately, their meaning in the present context, together with their advantages, remain unclear:

- The "bandwidth" advantage remains non-justified in the present context. In particular, the paragraph from lines 279 to 298 (of the redlined document) is confusing. Does wavelength multiplexing present an advantage in terms of parallelism? Or is it just a way of selectively tuning the weights (Fig. 3d)? By the way, Fig. 3d has been added in the new version, but no explanation is provided in the main text. Is this implementation instrumental for the device operation? I observe that, adding microrings to the photonic chip makes the architecture even more complex: hundreds of nanolasers, photonic cross-bars, amplifiers, balanced-detectors, and now even microrings... A realistic implementation of this can be extremely challenging, and even criticized. In my opinion, the authors should keep their designs realistic, at least they should clarify what are the fundamental ingredients needed for a proof-of-principle experimental demonstration.

- What do the authors mean by "latency"? Do they mean that extra delays in signal processing which could be alleviated in the photonic implementation? I note that the use of "latency" in the context of excitability might be confusing, because refractory times is a form of latency. The authors should clearly state what they mean by "latency" in this context and in what sense the nanolaser PSN are beneficial.

Additional remarks:

- Equation (3) is not written correctly: since the Wiener process dW_t is a differential quantity, the other terms should also be expressed as differentials (as in Equation (6)).

Reviewer #3

(Remarks to the Author)

The authors have adequately addressed all my comments.

One clarifying remark: In their reply about the "gain" becoming positive in Figure 2b, the authors discuss a "net gain", which includes the role of the amplifying and the absorbing section. That is now very clear to me but was initially not, perhaps for good reason: lines 225 and following state "Stimulated emission and absorption are described by a term G_S , where $G = \gamma r (2n_e - n_0)$ represents the gain, n_0 is the total number of dipoles, and γr is the radiative transition rate". This is the "gain" I had in mind while parsing figure 2b. Instead, in their discussion, the authors refer to another "gain", defined line 243 as " $G = \gamma r (2n_e - n_0) + \gamma r_a (2n_a - n_{0,a}) - \gamma$ is the net gain including photon damping". Just adding "net gain" instead of "gain" where applicable (in particular caption of fig 2) and perhaps choosing another form of "G" for the one defined line 225 and the one defined line 243 would remove the possibility of misunderstanding.

Version 2:

Reviewer comments:

Reviewer #2

(Remarks to the Author)

The authors have answered all my comments and questions. I can now recommend this manuscript for publication in Nature Communications.

Reviewer #1 (Remarks to the Author):

The manuscript proposes and analyzes the concept of spiking nanolasers, i.e. semiconductor lasers that emit ultrafast optical pulses, and demonstrates a model for computing based on sampling-based neural computation. The core idea is to use small photonic crystal nanolasers, each behaving as a stochastic spiking neuron, to perform Bayesian inference through sampling from learned probability distributions (e.g., Boltzmann machines). The authors provide a theoretical treatment comparing these nanolaser-based neurons to conventional leaky integrate-and-fire sampling (LIFS) neurons, show how to map typical Boltzmann machine parameters onto spiking nanolasers, and then illustrate the approach on generative tasks and pattern completion (e.g., small-scale MNIST).

Key contributions include:

- A derivation that links nanolaser gain dynamics and standard stochastic neuron models.
- Numerical experiments suggesting significant speed and potential power-efficiency gains over purely electronic spiking systems.
- Demonstrations of sampling accuracy for small toy problems (MNIST-like tasks).

Overall, the paper provides an interesting proposal for ultrafast, optically based neuromorphic systems. The results are timely, given the growing interest in photonic computing for machine learning, and the manuscript's focus on sampling-based models (rather than purely feedforward neural nets) highlights unique advantages of spiking neural substrates.

Authors' response: Thank you for this very encouraging assessment!

Significance and Relation to Established Literature

Optical spiking neurons could become a promising direction in neuromorphic engineering due to their potential for high-speed operation and parallel data transmission. Compared with other photonic neuromorphic proposals, the authors' emphasis on sampling, as opposed to deterministic classification, addresses an important niche. This aspect could be further empathized in the introduction.

Authors' response: Thank you for the suggestion. We have amended the introduction with several additional comments and links to recent and ongoing research (lines 65 ff., 75 ff. and 89 ff.).

The paper references prior demonstrations of spiking semiconductor lasers and places the present approach in context with existing neuromorphic devices like BrainScaleS and other hardware for spike-based sampling. Provided these speed and energy claims hold up in experimental settings, this work may have significant impact on the design of specialized photonic or hybrid photonic-electronic chips for Bayesian inference, optimization, and other tasks that benefit from fast random sampling. It builds upon established knowledge of Boltzmann machines and spiking neural networks but extends it into the realm of ultrafast photonic hardware.

Support for Conclusions and Claims

Spiking Mechanism & Neural Analogy: The theoretical equations linking the gain parameter of the laser to a membrane-potential-like stochastic variable appear consistent with known rate equations for semiconductor lasers. The authors' simulations further support that the excitability and refractoriness can mirror spiking neuron behavior.

Sampling Accuracy: Simulation results show that mapped Boltzmann parameters lead to correct sampling distributions in small networks, as well as decent performance on pattern completion for MNIST-like inputs. These results support the claim that photonic spiking neurons can implement sampling-based inference.

Energy and Performance Estimates: The authors provide speed estimates (on the order of gigahertz spiking rates), as well as approximate energy calculations that suggest orders-of-magnitude gains over purely electronic solutions. While these numbers are plausible, an experimental prototype or a more explicit breakdown of power budgets (e.g., how pumping, waveguide coupling, photodiode readouts, and any amplification circuitry scale with network size) would further validate the claims.

Authors’ response: Thank you for your suggestion. We now provide a detailed breakdown of power budgets in the “MNIST power consumption” section in the Methods and have amended the Discussion accordingly (line 655 ff.).

Additional Evidence: The paper convincingly motivates the theoretical viability of photonic sampling. However, real-world constraints, fabrication tolerances, optical losses, device mismatch, and large-scale routing complexities, might introduce deviations from idealized modeling. Some more in-depth discussion of those potential challenges, or references to relevant integration technologies, would strengthen the claims.

Authors’ response: It is true that our simulations did not explicitly account for fabrication tolerances. However, we expect the emission and absorption of spikes to be resistant against usual amounts of fixed-pattern noise. The former is explored extensively in [1], and the regions of parameters that provide excitable behavior are wide; fabrication procedures can be tuned accordingly. Spike reception is carried out by photodiodes, which are broadband and therefore resistant against spike wavelength variation.

While it is true that certain implementations of the coupling matrix can have issues, this does not pose a significant problem to our general approach. Therefore, we view this particular discussion as lying outside of the scope of our article.

Finally, in this work we assumed that the parameters of the system do not change over time. Therefore, the approach of calibration followed by training is sufficient. In practice, such parameter drifts are usually both weak and very slow, especially in comparison to the sampling time scales of PSN networks. Nevertheless, should this become necessary, it is straightforward to train or fine-tune the hardware system with the same wake-sleep algorithm as done in our simulations here. We expect such training to even improve the performance values reported here, as mentioned in the Discussion (line 621 ff.).

Your question about scaling is particularly interesting and relevant. While extremely difficult to answer comprehensively, we can extrapolate from the current state of the art in optical circuitry. Our PSN networks assume the existence of an optical circuit providing an all-to-all connection among the nanolasers. Ideally, nanolaser could be connected with free space optics and programmable arrays of micro-mirrors and spatial phase modulators [2, 3]; in this setting, a large number of channels can be controlled independently with a manageable level of losses. Spiking nanolasers such as those reported in [4] could be modified for free space emission using compact grating couplers or free-form optics with very low losses (~ 1 dB). On the other hand, MEMS technology offers a photonic integrated platform with ultra-low-loss switches connecting up to 240 ports [5]. The detailed breakdown of losses and required gain discussed in section “MNIST power consumption” is based on specifications given there. We note that the MEMS technology implemented there offers digital switching, which is unsuitable as such for our purposes. However, a continuous control of the weights is possible by using MEMS actuated directional couplers instead, with similar energy and loss specifications, as is the case for photonic matrices [6, 7], which feature 64 channels and are fully reconfigurable and low power. Finally, spectral multiplexing is possible owing to the incoherent nature of the interconnections, which could drastically increase the number of connections, as existing

multiplexing/demultiplexing technology based on weighting filter banks handles several spectral channels [8, 9].

A part of this answer has also been added to the end of the Discussion section (line 680 ff.).

Please address in the discussions points b, c, d and how these could be overcome and or limit the current work.

Potential Flaws and points that require revision

- a) Energy Estimation Details: The manuscript’s power estimates for large-scale networks could benefit from more specificity on amplifier or detector overheads, along with tolerance analysis of waveguide losses. As it stands, the rough estimates seem encouraging but may need tighter bounds or a more explicit breakdown.

Authors’ response: Regarding one specific detail of your comment, we note that nanoscale detectors have been shown not to need transimpedance amplifiers. In fact, their much reduced capacitance allows the use of large load resistance [10, 11]. Otherwise, we thank you for your suggestions and point to the more detailed power estimates provided now in the Discussion (line 655 ff.) and especially in the “MNIST power consumption” section of the Methods.

- b) Definition of Abbreviations: There are multiple instances where abbreviations appear before being defined. A comprehensive glossary or ensuring every acronym is defined upon first appearance would significantly improve clarity. (eg, SPS, PNS page 2 etc..)

Authors’ response: Already in the previous version, we tried to ensure that all acronyms are properly defined upon their first appearance, and every instance of every acronym also has a hyperlink pointing to its first appearance and definition. Upon rechecking, we found and amended a few missing hyperlinks and hope we haven’t missed anything this time. To improve readability, we have further eliminated some acronyms that only appeared infrequently in the manuscript (ANN, SBS and ISI).

- c) Methodological Completeness: While the theory and simulation approach are well described, readers from the photonic hardware community may want more details on device geometry or coupling schemes. Additional clarity about how well the authors expect the assumptions (e.g., linear summation of spikes, negligible inter-laser coherence) to hold in practice would help reproducibility.

Authors’ response: We have amended the Figure 1a and its caption to better represent the actual approach to coupling a PSN and a waveguide.

The nanolaser technology is described in detail in [12]. We provide this reference in the introduction (line 83 ff.). The spiking dynamics has been reported recently in a conference [4], and a manuscript is in preparation by our collaborators. We

address this work in the introduction as well (line 89 ff.). Our model is based on parameters extracted from measurements of these devices.

The point you raise about coherence is indeed very important. Coherent coupling between lasers could happen due to residual (unwanted) reflection in the switching matrix, although in principle this should be very low. Yet, we believe this effect could be made negligible as nanolasers will be designed to have a large spread of the emitted wavelengths. It is perfectly possible to engineer laser individually on the same chip, with the nanolaser technology described in [12]. The emitted frequencies could ideally cover the full C-band (~ 10 THz), while the spectral width of the emitted spikes is well below 100 GHz. This will also ensure linear summation of the spikes.

A part of this answer has also been added to the end of the Discussion section (line 710).

Methodology and Reproducibility

Soundness: The approach, using the known rate equations for semiconductor lasers augmented with noise terms, is standard and appears methodologically sound. Mapping Boltzmann parameters onto laser parameters follows from typical sampling-based spiking neuron theory.

Reproducibility: The methods section largely includes the relevant equations and standard parameter sets, enabling future researchers to replicate the simulations. Including a fully annotated list of parameters for each experiment (e.g., for the MNIST demonstration) and clarifying how the code is structured would enhance reproducibility.

Authors' response: We have added a “Simulation code details” section to the Methods clarifying the code structure and giving values of essential parameters.

In summary, the authors present a theoretically sound proposal. The authors make a strong case for how such ultrafast optical spiking neurons could perform sampling-based inference with notable improvements in speed and potentially power efficiency. Despite some points that would benefit from more details, particularly around practical power budgets and large-scale integration, the paper is overall a valuable contribution.

Authors' response: Thank you again for your encouraging comments, helpful suggestions and constructive criticism.

Recommendation: I recommend revisions to polish the manuscript, clarify energy estimates, ensure consistent definitions of abbreviations, and possibly strengthen the discussion around potential integration and scaling challenges.

Reviewer #2 (Remarks to the Author):

This work reports on the mapping from a photonic spiking neural network (PSN) to a Boltzmann machine (BM), and numerically demonstrates efficient computational tasks based on Bayesian inference. Among the myriad of today's optical computing approaches, this paper addresses the neuromorphic spiking response of semiconductor lasers with saturable absorbers, as a consequence of excitability. While excitability in semiconductor lasers is quite an old topic, intensively investigated in the past two decades, the advent of integrated photonic devices on the one hand, and the groundbreaking advancements in AI on the other hand, have boosted the research in this domain. In this respect the present manuscript represents quite a significant advancement. In particular, the small footprint and low consumption of photonic crystal nanolasers are claimed to offer a clear advantage over in-silico substrates, both in terms of speed and energy budget. Nevertheless, there are a few important issues that prevent me to positively recommend this work for publication in Nat. Comm. (points 1-4 below).

Authors' response: Thank you for your appreciation of our manuscript. Please find detailed replies to your reservations below, which we hope will alleviate your remaining concerns.

1) Broad audience and general interest. This manuscript is difficult to be understood by a non-specialist. It assumes a reader's background on Bayesian statistics and Boltzmann machines, which is far to be the case even in the community of optical neuromorphic computing. It is then of primary importance to introduce the main concepts of Bayesian inference, Boltzmann machines and sampling (both non-spiking and spiking), with a basic example (prior to spiking models) which would help the reader to a large extent. Otherwise, we might consider this work to be better suited to a more specialized journal (JCP or CPC). In my opinion, introducing these concepts is substantially more important than the details of the mapping ("Membrane potential of spiking nanolasers"): most of this subsection could be moved to a supplementary material, retaining only the fundamental result in the main text.

Authors' response: Accessibility is an important matter to us as well and we have thus taken your suggestions to heart. In particular, we have substantially extended our introduction to Boltzmann machines and sampling in the "Preliminaries" section (line 105 ff.) and to Bayesian statistics in the "Optical probabilistic inference" section (line 467 ff.).

2) Overall clarity. Many points remain obscure to me (many of them in connection to point 1). For instance: it is not clear how the training is performed, i.e. what parameters (weights?) are tuned, and how. Another example is the 2D representations of network states (Fig. 7.a and 8.e): the vast majority of readers ignore what t-SNE is, so a hint to interpret the representation axes is needed. Furthermore, sentences like "...the PSN network correctly samples from the prior" (l. 410) or "...by sampling from conditional probability distributions" (l. 345), or "the probability of each state sampled from the

inverse...” (l. 378) are very technical and specific from Bayesian statistics, so they need further clarification here. How PSP is implemented is not clear neither.

Authors’ response: As for the point above, we have taken your concerns to heart. We have revised and expanded multiple explanations of our methods, both in the main text and in the Methods. Beyond the already existing “Probabilistic inference training” section, we have now added detailed descriptions of the following procedures to the Methods:

- the contrastive divergence method used in training to sample from arbitrary distributions (section “Contrastive divergence”, line 828 ff.);
- the creation of the star plots used for visualization of probabilistic inference (section “Probabilistic inference visualization”, line 867 ff.);
- t-Stochastic Neighbor Embedding (t-SNE) used for visualization of the guided dreaming results (section “t-Stochastic Neighbor Embedding”, line 1010 ff.).

As for the other points, we have revised the surrounding text; together with the additional information in the Methods (see above), we hope that these explanations are clearer now:

- on “sampling from prior”, see line 491 ff.;
- on “by sampling from conditional probability distributions”, see line 411 ff.;
- on “sampled from the inverse...”, see line 445 ff.;
- on the PSP: the explanation was given in the caption of Figure 3, which, we concur, might be missed by readers; we have thus added a short explanation and a reference in the main text (line 310 ff.);
- for more clarifications and explanations, see also lines 344 ff. and 481 ff.

3) The optical spiking advantage. It is claimed that the spiking dynamics is advantageous both in terms of energy saving and robustness to noise. While I acknowledge this as general statement, which may hold true in neuroscience, it is not clear to what extent it applies to the particular photonic system studied here. Specifically:

- a. Power. CW laser-threshold powers are even smaller than those of pulsed devices.

Authors’ response: We agree with the argument that all else being equal, a saturable absorber increases losses and threshold pumping as a result. That said, we do not believe the argument is applicable in the context of this work: CW lasers would not be suitable for stochastic neural networks due to a lack of prominent stochasticity.

- b. Bandwidth. Frequency multiplexing is not clear in the context of the proposed photonic architecture, and should be commented.

Authors' response: We have added an explanation after the multiplexing is mentioned (line 286 ff.).

- c. Losses. Interconnect matrix is based on waveguide crossbar arrays. Then the scaling of crossing losses is a potential issue in large networks, which is not discussed.

Authors' response: We agree with this point, which was also brought up by the Reviewer #1. In general, losses can be compensated for by associating each port (each of them grouping lasers emitting in separated spectral channels) with a nanoscale semiconductor optical amplifier (i.e., based on the same technology) providing gain yet with very low power requirements (e.g., raising the average power emitted by the laser from 1 μ W to 100 μ W). For many further details, we kindly refer to our responses to Reviewer #1, sections "Additional Evidence", "Energy Estimation Details" and "Methodological Completeness".

4) The context of excitability in lasers. The references in the Introduction do not fairly reflect the historical developments. The first cited works on excitability (Refs. [23-25], Wünsch et al., Romeira et al., Prucnal et al.) do not accurately retrace the seminal studies on optical excitability. The first report on excitability was by F. Plaza et al. (EPL 38, 85, 1997) in a CO₂ laser with SA, followed by experiments in semiconductor lasers (M. Giudici et al., Phys. Rev. E 55, 6414, 1997) and Nd-YAG lasers with SA (Larotonda et al., PRA 65, 033812, 2002). In the context of photonic crystals, the first demonstration in a nanocavity was reported in Brunstein et al, PRA 85, 031803 (2012).

Authors' response: We thank you for sharing these references, which we have now added to the Introduction (line 75 ff.).

Minor comments:

- i. Abstract. Saying that speed is an advantage of optics w.r.t electronics is not true in general, as an electronic signal also propagates at the speed of light in metallic wires. In addition, in the abstract (l. 5), what do authors mean by "conventional semiconductor devices"?

Authors' response: We agree with this point, as "speed" is a rather unspecific term that may lead to confusion. Rather, we meant "bandwidth" and/or "latency", that now replace "speed" where appropriate. As for "conventional semiconductor devices": we meant "digital electronics". See our amendments in lines 2 ff. and 5 ff. in the Abstract and 62 ff. and 65 ff. in the Introduction.

- ii. Possible confusion in the notation: β is used as spont. emission factor of nanolasers, and as a parameter for probability distributions.

Authors' response: Good catch! We have removed β as the spontaneous emission factor (line 204 ff.) as it is not used elsewhere.

- iii. Absolute/relative refractoriness are assumed to be well-known concepts, yet they're most likely ignored by the general reader.

Authors' response: Agreed, we have added a brief explanation in the Discussion (line 605 ff.).

- iv. Line 264 should read: "Second, changes of n_e due to incoming spikes need to be small. . ."

Authors' response: Good catch, thank you (line 330 ff.).

Reviewer #3 (Remarks to the Author):

The manuscript demonstrates the feasibility of general spike-based sampling using a network of spiking lasers and discuss several examples of this sampling operation for the realization of complex machine learning tasks. The work is very complete, very well written and should be interesting to a large community ranging from machine learning specialists to photonics experts. The results include foundational points such as how to map the spiking probability to BM weights and very advanced tasks such as "guided dreaming". For the completeness of the work, its novelty, informative content and general quality, I warmly recommend publication of the manuscript.

Authors' response: Thank you for your very encouraging assessment of our work!

However, a few points deserve clarification in my opinion:

- The coupling between nodes is considered to take place via embedded photodetectors, aiming at incoherent transmission of spikes. Is coupling through saturation of the absorber section impossible or impractical?

Authors' response: We believe it is possible, but impractical as it would require the coupling of a spike inside the receiving nanolaser. A sufficient coupling requires the alignment of resonances, which is challenging for a large number of nanolasers.

- The model uses variable S to describe a photon number, which must therefore be positive. What prevents S from becoming negative in presence of the stochastic term F ? This is particularly relevant considering that most of the time the system is not emitting a spike thus in absence of noise $S=0$ (figure 1b).

Authors' response: Indeed, noise can move the variables outside the "physical" range of values. During our simulations, S was forced to always be above a small positive number, and n_e and n_a to always be at least 0. However, this way we may have significantly altered the dynamics of the system. For this reason, we compared the continuous model to its discrete counterpart which does not suffer from such issues in the "Discrete nanolaser model" section of Methods, which now also includes this discussion (line 802 ff.). We found that both models give very close results.

- Figure 2b shows time traces of the normalized gain, which is, as expected mostly slightly below 0. However, at the onset of spikes, the gain depletion is anticipated by a very large increase of the gain (particularly visible for instance on spikes 3,4,5), which takes place at a time scale which is comparable to the following depletion (which is mediated by the photon number growth). What is the origin of this very sharp increase? Could it be related to the previous point?

Authors’ response: The sharp increase is due to a positive feedback loop created by the saturable absorber (SA). The increase of the photon count due to the spontaneous emission reduces the electron density in the pumped section, which, in turn, reduces its gain. However, in our structure, the SA is a nanolaser itself, which shares the pool of photons with the pumped section. The SA will absorb the newborn photons to create free electrons. As the SA is not pumped, its differential gain is higher than in the pumped section, so the change of gain in the SA is positive and its magnitude is higher than the loss of gain in the pumped section. As a result, the overall gain of the structure has increased, which will create more photons, which will further increase the gain – a positive feedback loop. We observe this behaviour with the discrete model as well, therefore it is unrelated to noise-induced $S < 0$.

As for the “particularly visible ... on spikes 3,4,5” it is an unfortunate outcome of insufficient density of points in the figure: some gain peaks were lost between the subsequent points. We have increased the density of points around the spikes, and all spikes look similar now. We have also added a comparison of gain timetraces in the “Discrete nanolaser model” section (Figure 9) in the Methods.

- The “wake sleep” algorithm **must** be briefly described, with a reference for more information, if the work is to be readable also to non-experts in stochastic neural networks training.

Authors’ response: We absolutely agree. The explanation of the wake-sleep algorithm can be found in the “Probabilistic inference training” section in the Methods. However, we had unfortunately omitted a corresponding reference in the main text, which is now given at the first occurrence of the term (line 489 ff.).

Points to be considered: The “optical probabilistic inference” and “learning from data” sections seemed to me as “logical” specific applications of the section “optical sampling from arbitrary distribution”. If it is the case, I fear these sections do not really help the reader. I understand they constitute “bright” points and their implementation is probably worth a description but if they are really direct consequences of “optical sampling from arbitrary distribution” the reader would benefit from being informed of that very clearly.

Authors’ response: We have tried to better motivate our progression of functional applications the end of the “Membrane potential of spiking nanolasers” section (line 377 ff.).

Minor points:

- figure 1a: is a spike really shorter than the laser resonator as suggested by fig 1a? If yes, a PDE system would really be required to describe the physics. If not, then a single mode model as used here is fine but then figure 1a is a bit misleading.

Authors' response: This is quite a subtle point, you are absolutely right and thank you for pointing it out. The spikes are indeed actually “long”, as their FWHM of ~ 30 ps corresponds to ~ 3 mm, whereas a typical photonic crystal is roughly $10 \mu\text{m}$ long. We have therefore removed the spikes from the Figure 1a, as they were also not very instructive at that point in the first place.

- “alpha shaped” is used here and there in the manuscript but not defined

Authors' response: We have added a definition at the first mention (line 313 ff.).

- The connection between BMs and spiking networks cites 11-13, I believe Hinton, G. E., & Brown, A. (1999). Spiking boltzmann machines. *Advances in neural information processing systems*, 12. should be credited.

Authors' response: Thank you for bringing our attention to this article. While the proposed implementation is quite different from ours, we were very happy to include its citation in the manuscript (line 46 ff.).

References

1. Delmulle, M. *Le nanolaser neurone: excitabilité dans les lasers à cristaux photoniques* PhD thesis (2023).
2. Chen, Z. *et al.* Deep learning with coherent VCSEL neural networks. *Nature Photonics* **17**, 723–730 (2023).
3. Pflüger, M. *et al.* Experimental reservoir computing with diffractively coupled VCSELs. *Optics Letters* **49**, 2285–2288 (2024).
4. Delmulle, M *et al.* *Excitability in a PhC nanolaser with an integrated saturable absorber* in *European Quantum Electronics Conference* (2023), jsiii_6-2.
5. Seok, T. J., Kwon, K., Henriksson, J., Luo, J. & Wu, M. C. Wafer-scale silicon photonic switches beyond die size limit. *Optica* **6**, 490–494 (2019).
6. Hua, S. *et al.* An integrated large-scale photonic accelerator with ultralow latency. *Nature* **640**, 361–367 (2025).
7. Ahmed, S. R. *et al.* Universal photonic artificial intelligence acceleration. *Nature* **640**, 368–374 (2025).
8. Ohno, S., Tang, R., Toprasertpong, K., Takagi, S. & Takenaka, M. Si microring resonator crossbar array for on-chip inference and training of the optical neural network. *Acs Photonics* **9**, 2614–2622 (2022).

9. Tait, A. N. *et al.* Neuromorphic photonic networks using silicon photonic weight banks. *Scientific reports* **7**, 7430 (2017).
10. Nozaki, K. *et al.* Photonic-crystal nano-photodetector with ultrasmall capacitance for on-chip light-to-voltage conversion without an amplifier. *Optica* **3**, 483–492 (2016).
11. Nozaki, K. *et al.* Femtofarad optoelectronic integration demonstrating energy-saving signal conversion and nonlinear functions. *Nature Photonics* **13**, 454–459 (2019).
12. Crosnier, G. *et al.* Hybrid indium phosphide-on-silicon nanolaser diode. *Nature Photonics* **11**, 297–300 (2017).

Sincerely,

Ivan K. Boikov, Alfredo de Rossi and Mihai A. Petrovici

Reviewer #1 (Remarks to the Author):

I thank the authors for thoroughly addressing all of the substantive points raised: they have strengthened the introduction by emphasizing the sampling niche with new citations; provided a detailed “MNIST power consumption” section to clarify energy budgets; discussed fabrication tolerances and routing challenges by covering noise tolerance, photodiode resilience, and options such as free-space/MEMS/spectral multiplexing; cleaned up infrequent acronyms, fixed hyperlink definitions, and added a “Simulation code details” section with annotated parameters to improve methodological clarity and reproducibility.

A single follow-up question remain: do the authors intend to publicly release their simulation code, and how should readers interpret the statement “Simulation code is available upon a reasonable request”? Could the authors clarify under what conditions and through what process one might request and obtain the code?

Authors’ response: While the manuscript offers a full and detailed description of the performed simulations, we are open to receiving individual requests for direct access to the code. We will meet any such request with our full support, but our institutional regulations require us to obtain case-by-case approval by the management of Thales.

Reviewer #2 (Remarks to the Author):

Thank you for having modified the manuscript according to my comments and questions. Most of the “obscure points” have now been clarified in the text. Still there are a couple of issues that need to be addressed:

1. The “optical spiking advantage”. The authors’ response to my point 3a) is: “we do not believe the argument is applicable in the context of this work: CW lasers would not be suitable for stochastic neural networks due to a lack of prominent stochasticity.”. This is not true for a nanolaser: close enough to laser threshold, the amplified spontaneous emission noise dominates the dynamics. Therefore, the question remains open: are spikes in optical NNs advantageous enough –as they are in the brain–, justifying its implementation over other (stochastic) approaches in a photonic context? I think a short discussion on this point is needed.

Authors’ response: We agree that our previous response about CW lasers not being suitable for this task was not nuanced enough. We definitely do not mean to claim that our specific implementation of Boltzmann sampling is the only way to perform stochastic Bayesian computation. The important point here is that the solution presented in our manuscript represents a near-perfect match for neural sampling as a demonstrated means of carrying out sampling from Boltzmann distributions. This framework relies crucially on stochastic, pulsed neuronal responses followed by pronounced refractory periods. We have amended the introduction (see line 92 ff.) to hopefully clarify this point better.

2. The authors have revised their use of the word “speed”, and now focus on “bandwidth” and/or “latency”. Unfortunately, their meaning in the present context, together with their advantages, remain unclear:

- The “bandwidth” advantage remains non-justified in the present context. In particular, the paragraph from lines 279 to 298 (of the redlined document) is confusing. Does wavelength multiplexing present an advantage in terms of parallelism? Or is it just a way of selectively tuning the weights (Fig. 3d)?. By the way, Fig. 3d has been added in the new version, but no explanation is provided in the main text. Is this implementation instrumental for the device operation? I observe that, adding microrings to the photonic chip makes the architecture even more complex: hundreds of nanolasers, photonic cross-bars, amplifiers, balanced-detectors, and now even microrings... A realistic implementation of this can be extremely challenging, and even criticized. In my opinion, the authors should keep their designs realistic, at least they should clarify what are the fundamental ingredients needed for a proof-of-principle experimental demonstration.

Authors’ response: We believe the bandwidth advantage is justified, but not necessarily in the sense of handling faster inputs, but facilitating data transfer between internal components. The broad optical bandwidth allows

for wavelength multiplexing which makes the coupling between nanolasers more compact without losing data transfer parallelism. For example, in digital spiking networks, several neurons share the same bus; each neuron must wait until the bus is free to perform I/O. In optics, one waveguide can transfer spikes from multiple nanolasers, provided their resonances are separated.

Wavelength multiplexing is not required for tuning the weights of nanolaser coupling. If spikes from different nanolasers interfere (which is a challenge in itself), one might consider a matrix-vector multiplication circuit such as in [Shen et al., Nat. Photon. (2017)].

We agree that the technology is extremely challenging, although integrated photonics is improving steadily and in the examples shown in the cited references, photonic chips with several hundreds of functional components exist already. We do not claim that our approach is “better” than others and the experimental implementation we suggest is by no means expected to be optimal, rather, we regard it as an educated guess of a feasible architecture. We have now addressed this point explicitly on line 266 ff. and added references to Figures 3c,d in the discussions that follow.

- What do the authors mean by “latency”? Do they mean that extra delays in signal processing which could be alleviated in the photonic implementation? I note that the use of “latency” in the context of excitability might be confusing, because refractory times is a form of latency. The authors should clearly state what they mean by “latency” in this context and in what sense the nanolaser PSN are beneficial.

Authors’ response: Indeed, latency can be interpreted differently depending on the context. In our text, latency only appears in the introduction, in the context of computing speed, as also addressed by the provided references [23-26]. We modified the sentence immediately following the first appearance of the term (line 61 ff.) to make this even clearer. In other words, “low latency” can also be translated to “short time-to-solution”. As we show later on, the sampling speeds achieved by our proposed methods are orders of magnitude higher than those of state-of-the-art electronic neuromorphic systems, translating to a corresponding decrease in the time required to obtain an approximation of the sought probability distribution with some desired degree of precision.

Additional remarks:

- Equation (3) is not written correctly: since the Wiener process dW_t is a differential quantity, the other terms should also be expressed as differentials (as in Equation (6)).

Authors’ response: Good catch, thank you! We have amended the equation.

Reviewer #3 (Remarks to the Author):

The authors have adequately addressed all my comments.

One clarifying remark: In their reply about the “gain” becoming positive in Figure 2b, the authors discuss a “net gain”, which includes the role of the amplifying and the absorbing section. That is now very clear to me but was initially not, perhaps for good reason: lines 225 and following state “Stimulated emission and absorption are described by a term GS , where $G = \gamma r(2n_e - n_0)$ represents the gain, n_0 is the total number of dipoles, and γ_r is the radiative transition rate”. This is the “gain” I had in mind while parsing figure 2b. Instead, in their discussion, the authors refer to another “gain”, defined line 243 as “ $G = \gamma_r(2n_e - n_0) + \gamma_{r,a}(2n_a - n_{0,a}) - \gamma$ is the net gain including photon damping”. Just adding “net gain” instead of “gain” where applicable (in particular caption of fig 2) and perhaps choosing another form of “G” for the one defined line 225 and the one defined line 243 would remove the possibility of misunderstanding.

Authors’ response: That is true, thank you! We now refer to $G = \gamma_r(2n_e - n_0) + \gamma_{r,a}(2n_a - n_{0,a}) - \gamma$ as “gain”. Please see lines 217 and 236 ff.

Sincerely,

Ivan K. Boikov, Alfredo de Rossi and Mihai A. Petrovici